# Variational Multi-Task Learning with Gumbel-Softmax Priors

**Jiayi Shen**[1], **Xiantong Zhen**[1,2], **Marcel Worring**[1], **Ling Shao**[2]
[1]AIM Lab, University of Amsterdam, Netherlands
[2]Inception Institute of Artificial Intelligence, Abu Dhabi, UAE

## Abstract

Multi-task learning aims to explore task relatedness to improve individual tasks, which is of particular significance in the challenging scenario that only limited data is available for each task. To tackle this challenge, we propose variational multi-task learning (VMTL), a general probabilistic inference framework for learning multiple related tasks. We cast multi-task learning as a variational Bayesian inference problem, in which task relatedness is explored in a unified manner by specifying priors. To incorporate shared knowledge into each task, we design the prior of a task to be a learnable mixture of the variational posteriors of other related tasks, which is learned by the Gumbel-Softmax technique. In contrast to previous methods, our VMTL can exploit task relatedness for both representations and classifiers in a principled way by jointly inferring their posteriors. This enables individual tasks to fully leverage inductive biases provided by related tasks, therefore improving the overall performance of all tasks. Experimental results demonstrate that the proposed VMTL is able to effectively tackle a variety of challenging multi-task learning settings with limited training data for both classification and regression. Our method consistently surpasses previous methods, including strong Bayesian approaches, and achieves state-of-the-art performance on five benchmark datasets.

## 1 Introduction

Multi-task learning [8] exploits knowledge shared among related tasks to improve their overall performance. This is especially significant when only limited data is available for each task. The central goal in multi-task learning is to explore task relatedness to improve each individual task [2, 54], which is non-trivial since the underlying relationships among tasks can be complicated and highly nonlinear [52]. Early works deal with this by learning shared features, designing regularizers imposed on parameters [37, 24, 23, 21] or exploring priors over parameters [20, 3, 48, 53, 54]. Recently, multi-task deep neural networks have been developed, learning shared representations in the feature layers while keeping classifier layers independent [33, 38, 55, 31]. Some methods [32, 25, 9, 14] learn a small number of related tasks based on one shared input. However, lots of training data is usually available. Few-shot learning [12] addresses multi-task learning with limited data, but usually relies on a large number of related tasks. Further, in general, most existing works explore either representations or classifiers for shared knowledge, leaving exploring them jointly an open problem.

In this work, we tackle the following challenging multi-task learning setting [33]: each task contains very limited training data and only a handful of related tasks are available to gain shared knowledge from. In addition, instead of sharing the same input, each of the multiple tasks has its own input space from a different domain and these tasks are related by sharing the same target space, e.g., the same class labels in classification tasks. In this setting, it is difficult to learn a proper model for each task independently without overfitting [33, 52]. It is therefore crucial to leverage the inductive bias [4] provided by related tasks learned simultaneously.

To address this, we propose variational multi-task learning (VMTL), a novel variational Bayesian inference approach that can explore task relatedness in a unified way. Specifically, as shown in Fig. 1, we develop a probabilistic latent variable model by treating both the feature representation and the classifier as latent variables. To explore task relatedness, we place conditional priors over the latent variables, which are dependent on data from other related tasks. By doing so, multi-task learning is cast as the joint variational inference of feature representations and classifiers for all tasks in a single unified framework. The probabilistic framework enables the model to better capture the uncertainty caused by limited data in each task [13].

To leverage the knowledge provided by related tasks, we propose to specify the priors of each task as a mixture of the variational posteriors of other tasks. In particular, the mixing weights are constructed with the Gumbel-Softmax technique [22] and jointly learned with the probabilistic modeling parameters by back-propagation. The obtained Gumbel-Softmax priors enable the model to effectively explore different correlation patterns among tasks and, more importantly, provide a unified way to explore shared knowledge jointly for representations and classifiers. We validate the effectiveness of the proposed VMTL by extensive evaluation on five multi-task learning benchmarks with limited data for both classification and regression. The results demonstrate the benefit of variational Bayesian inference for modeling multi-task learning. VMTL consistently achieves state-of-the-art performance and surpasses previous methods on all tasks.

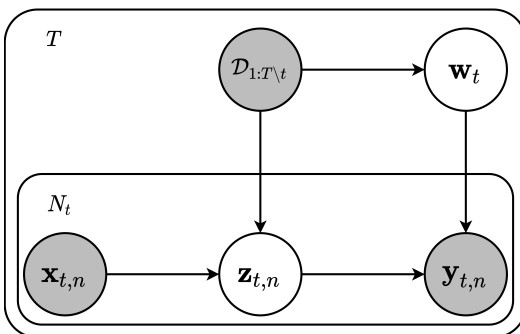

Fig. 1: A graphical illustration of the proposed model for multi-task learning. $(\mathbf{x}_{t,n}, \mathbf{y}_{t,n})$ is the $n$-th sample from task $t$. $\mathbf{w}_t$ and $\mathbf{z}_{t,n}$ are the introduced latent variables, i.e., a classifier/regressor and representation, respectively, whose priors are conditioned on the data from other tasks, $\mathcal{D}_{1:T \backslash t}$.

## 2 Methodology

Consider a set of $T$ related tasks, each of which is a classification or regression problem. Tasks in this paper share the same label space for classification or the same target space for regression but have different distributions. Each task $t$ has its own training data $\mathcal{D}_t = \{\mathbf{x}_{t,n}, \mathbf{y}_{t,n}\}_{n=1}^{N_t}$, where $N_t$ is the number of training samples. We consider the challenging setting where each task has limited labeled data, which makes it difficult to learn a proper model for each task independently [33, 52]. In contrast to most previous works, we will explore the knowledge shared among tasks for learning both the classifiers/regressors and representations of each task. Note that in this section we derive our methodology mainly using terminologies related to classification tasks, but it is also applicable to regression tasks.

### 2.1 Variational Multi-Task Learning

To better capture uncertainty caused by limited data, we explore multi-task learning in a probabilistic framework. We define the conditional predictive distribution: $p(\mathcal{Y}_t | \mathcal{X}_t, \mathcal{D}_{1:T \backslash t})$ with respect to the current task $\mathcal{D}_t$: $\mathcal{Y}_t = \{\mathbf{y}_{t,n}\}_{n=1}^{N_t}$ and $\mathcal{X}_t = \{\mathbf{x}_{t,n}\}_{n=1}^{N_t}$, where we use $\mathcal{D}_{1:T \backslash t}$ to denote the data from all tasks excluding $\mathcal{D}_t$ of task $t$. From a probabilistic perspective, jointly learning multiple tasks amounts to maximizing the conditional predictive log-likelihood as follows:

$$\frac{1}{T} \sum_{t=1}^{T} \log p(\mathcal{Y}_t | \mathcal{X}_t, \mathcal{D}_{1:T \backslash t}) = \frac{1}{T} \sum_{t=1}^{T} \sum_{n=1}^{N_t} \log p(\mathbf{y}_{t,n} | \mathbf{x}_{t,n}, \mathcal{D}_{1:T \backslash t}). \tag{1}$$

We condition the prediction for samples in each task on the data of other tasks, $\mathcal{D}_{1:T \backslash t}$, to leverage the shared knowledge from related tasks.

The probabilistic multi-task learning formalism in (1) provides a general framework to explore task relatedness by conditioning on other tasks. More importantly, it enables the model to incorporate shared knowledge from related tasks into the learning of both the classifier and the feature representation in a unified way by specifying their priors.

We introduce the latent variables $\mathbf{w}_t$ and $\mathbf{z}_{t,n}$ to represent the classifier and the latent representation of the sample $\mathbf{x}_{t,n}$, respectively. The joint distribution for the task $t$ can be factorized as

$$p(\mathcal{Y}_t, \mathcal{Z}_t, \mathbf{w}_t | \mathcal{X}_t, \mathcal{D}_{1:T \setminus t}) = \prod_{n=1}^{N_t} p(\mathbf{y}_{t,n} | \mathbf{z}_{t,n}, \mathbf{w}_t) p(\mathbf{z}_{t,n} | \mathbf{x}_{t,n}, \mathcal{D}_{1:T \setminus t}) p(\mathbf{w}_t | \mathcal{D}_{1:T \setminus t}), \qquad (2)$$

where we use $\mathcal{Z}_t = \{\mathbf{z}_{t,n}\}_{n=1}^{N_t}$ to collectively represent the set of latent representations of samples in task $t$. Here, we specify conditional priors dependent on other tasks to explore task relatedness. The probabilistic latent variables capture uncertainties at different levels: $\mathbf{z}_{t,n}$ captures the uncertainty of each sample, while $\mathbf{w}_t$ works at the categorical level. A graphical illustration of the probabilistic latent variable model is shown in Fig. 1.

Solving the model for multi-task learning involves inferring the true joint posterior $p(\mathcal{Z}_t, \mathbf{w}_t | \mathcal{D}_{1:T})$ over all latent variables $\mathcal{Z}_t$ and $\mathbf{w}_t$, which is generally intractable. We introduce a variational joint distribution $q(\mathcal{Z}_t, \mathbf{w}_t | \mathcal{D}_t)$ for the current task to approximate the true posterior. Also under the conditional independence assumption, the joint variational posterior distribution can be factorized with respect to classifiers and latent representations as follows:

$$q(\mathcal{Z}_t, \mathbf{w}_t | \mathcal{D}_t) = q_\theta(\mathbf{w}_t | \mathcal{D}_t) \prod_{n=1}^{N_t} q_\phi(\mathbf{z}_{t,n} | \mathbf{x}_{t,n}), \qquad (3)$$

where $q_\theta(\mathbf{w}_t | \mathcal{D}_t)$ and $q_\phi(\mathbf{z}_{t,n} | \mathbf{x}_{t,n})$ are variational posterior distributions for classifiers and latent representations respectively, and $\theta$ and $\phi$ are the associated parameters. For computational feasibility, they are defined as fully factorized Gaussians, as is common practice [28, 5].

To obtain a good approximation, the variational posterior should be close to the true posterior. This is usually achieved by minimizing the Kullback-Leibler (KL) divergence between them:

$$\mathbb{D}_{\mathrm{KL}}\big[q(\mathcal{Z}_t, \mathbf{w}_t | \mathcal{D}_t) || p(\mathcal{Z}_t, \mathbf{w}_t | \mathcal{D}_{1:T})\big]. \qquad (4)$$

By applying Bayes' rule to the true posterior, we derive the evidence lower-bound (ELBO) with the probabilistic classifier and representation for the conditional predictive log-likelihood in (1) :

$$\frac{1}{T} \sum_{t=1}^{T} \log p(\mathcal{Y}_t | \mathcal{X}_t, \mathcal{D}_{1:T \setminus t}) \geq \frac{1}{T} \sum_{t=1}^{T} \bigg\{ \sum_{n=1}^{N_t} \Big\{ \mathbb{E}_{\mathbf{w}_t \sim q_\theta} \mathbb{E}_{\mathbf{z}_{t,n} \sim q_\phi} [\log p(\mathbf{y}_{t,n} | \mathbf{z}_{t,n}, \mathbf{w}_t)]$$
$$- \mathbb{D}_{\mathrm{KL}}[q_\phi(\mathbf{z}_{t,n} | \mathbf{x}_{t,n}) || p(\mathbf{z}_{t,n} | \mathbf{x}_{t,n}, \mathcal{D}_{1:T \setminus t})] \Big\} - \mathbb{D}_{\mathrm{KL}}\big[q_\theta(\mathbf{w}_t | \mathcal{D}_t) || p(\mathbf{w}_t | \mathcal{D}_{1:T \setminus t})\big] \bigg\}. \qquad (5)$$

The objective function provides a general probabilistic inference framework for multi-task learning. As priors naturally serve as regularizations in Bayesian inference, they offer a unified way of sharing information across multiple tasks for improving both classifiers and representations. The detailed derivation is given in the supplementary materials.

In what follows, we will describe the specification of the prior distributions over the classifier and the representation, as well as their variational posterior distributions.

## 2.2 Learning Task Relatedness via Gumbel-Softmax Priors

The proposed variational multi-task inference framework enables related tasks to provide supportive information for the current task through the conditional priors of the latent variables. It offers a unified way to incorporate shared knowledge from related tasks into the inference of representations and classifiers for individual tasks.

**Classifiers**  To leverage the shared knowledge, we propose to specify the prior of the classifier for the current task using the variational posteriors over classifiers of other tasks:

$$p(\mathbf{w}_t^{(\eta)} | \mathcal{D}_{1:T \setminus t}) := \sum_{i \neq t} \alpha_{ti} q_\theta(\mathbf{w}_i^{(\eta-1)} | \mathcal{D}_i), \qquad (6)$$

where $\eta$ denotes the $\eta$-th iteration. In practice, to avoid entanglement among tasks, we define the prior of task $t$ in the current iteration to be a combination of the variational posteriors [40] of the remaining

tasks from the last iteration. Particularly, our designed prior resembles the empirical Bayesian prior [20, 3, 46, 44] in that the prior of each task is specified by the observed data of other tasks, which provides a principled way to explore the shared knowledge among multiple tasks.

During training, we aim to have each task learn more shared knowledge from its most relevant tasks, while maximally reducing the interference from irrelevant tasks. To this end, we adopt the Gumbel-Softmax technique learn the mixing weights for all related tasks and define the mixing weights as follows:

$$\alpha_{ti} = \frac{\exp((\log \pi_{ti} + g_{ti})/\tau)}{\sum_{i \neq t} \exp((\log \pi_{ti} + g_{ti})/\tau))}. \tag{7}$$

Here $\alpha_{ti}$ is the mixing weight that indicates the relatedness between tasks $t$ and $i$. $\pi_{ti}$ is the learnable parameter in the Gumbel-Softmax formulation, which denotes the probability of two tasks transferring positive knowledge. $g_{ti}$ is sampled from a Gumbel distribution, using inverse transform sampling by drawing $u \sim \text{Uniform}(0, 1)$ and computing $g = -\log(-\log(u))$. $\tau$ is the softmax temperature. By using the Gumbel-Softmax technique, our model effectively handles possible negative transfer between tasks. The Gumbel-Softmax technique encourages the model to reduce the interference from the less relevant tasks by minimizing the corresponding mixing weights. The more negative the effects of interference between pairwise tasks are, the smaller the mixing weight is likely to be.

With respect to the inference of the variational posteriors, we define them as fully factorized Gaussians for each class independently:

$$q_\theta(\mathbf{w}_t|\mathcal{D}_t) = \prod_{c=1}^{C} q_\theta(\mathbf{w}_{t,c}|\mathcal{D}_{t,c}) = \prod_{c=1}^{C} \mathcal{N}(\boldsymbol{\mu}_{t,c}, \text{diag}(\boldsymbol{\sigma}_{t,c}^2)). \tag{8}$$

where $\boldsymbol{\mu}_{t,c}$ and $\boldsymbol{\sigma}_{t,c}^2$ can be directly learned with back-propagation [5].

As an alternative to direct inference, classifiers can also be inferred by the amortization technique [17], which will further reduce the computational cost. In particular, different classes share the inference network to generate the parameters of the specific classifier. In practice, the inference network takes the mean of the feature representations in each class as input and returns the parameters $\boldsymbol{\mu}_{t,c}$ and $\boldsymbol{\sigma}_{t,c}$ for $\mathbf{w}_{t,c}$. The amortized classifier inference enables the cost to be shared across classes, which reduces the overall cost. Therefore, it offers an effective way to handle scenarios with a large number of object classes and can still produce competitive performance, as shown in our experiments, even in the presence of the amortization gap [10].

**Representations** In a similar way to (6), we specify the prior over the latent representation $\mathbf{z}_{t,n}$ of a sample $\mathbf{x}_{t,n}$ as a mixture of distributions conditioned on the data of every other task, as follows:

$$p(\mathbf{z}_{t,n}^{(\eta)}|\mathbf{x}_{t,n}, \mathcal{D}_{1:T \setminus t}) := \sum_{i \neq t} \beta_{ti} q_\phi(\mathbf{z}_{t,n}^{(\eta-1)}|\mathbf{x}_{t,n}, \mathcal{D}_i). \tag{9}$$

Here, $\beta$ is the mixing weight which is defined in a similar way as in (7). The conditional distribution $q_\phi(\mathbf{z}_{t,n}|\mathbf{x}_{t,n}, \mathcal{D}_i)$ on the right hand side of (9) indicates that we leverage the data $\mathcal{D}_i$ from every other task $i$ to help the model infer the latent representation of $\mathbf{x}_{t,n}$. The contribution of each other task is determined by the learnable mixing weight. In practice, the distribution $q_\phi(\mathbf{z}_{t,n}|\mathbf{x}_{t,n}, \mathcal{D}_i)$ is inferred by an amortized network [15, 28]. To be more specific, the inference network takes an aggregated representation of $\mathbf{x}_{t,n}$ and $\mathcal{D}_i$ as input and returns the parameters of distribution $q_\phi$. The aggregated representation is formulated as (10) which is established by the cross attention mechanism [26]. The sample $\mathbf{x}_{t,n}$ from the current task acts as a query, and $D_{i,c}$ plays the roles of the key and the value. $D_{i,c}$ includes samples from the $i$-th task, which have the same label as $\mathbf{x}_{t,n}$. This is formulated as:

$$f(\mathbf{x}_{t,n}, \mathcal{D}_i) = \text{DotProduct}(\mathbf{x}_{t,n}, D_{i,c}, D_{i,c}) := \text{softmax}(\frac{\mathbf{x}_{t,n} D_{i,c}^\top}{\sqrt{d}}) D_{i,c}, \tag{10}$$

where $f$ is the aggregation function, and $D_{i,c}$ is a matrix, with each row containing a sample from class $c$ of the $i$-th related task. $d$ is the dimension of the input feature. Since we are dealing with supervised learning in this work, class labels of training samples are always available at training time. The intuition here is to find similar samples to help build the representation of the current sample.

The inference of the variational posterior $q_\phi(\mathbf{z}_{t,n}|\mathbf{x}_{t,n})$ over latent representations is also achieved using the amortization technique [15, 28]. The amortized inference network takes $\mathbf{x}_{t,n}$ as input and returns the statistics of its probabilistic latent representation.

## 2.3 Empirical Objective Function

By integrating (6) and (9) into (5), we obtain the following empirical objective for variational multi-task learning with Gumbel-Softmax priors:

$$\hat{\mathcal{L}}_{\text{VMTL}}(\theta, \phi, \alpha, \beta) = \frac{1}{T} \sum_{t=1}^{T} \left\{ \sum_{n=1}^{N_t} \left\{ \frac{1}{ML} \sum_{\ell=1}^{L} \sum_{m=1}^{M} \left[ -\log p(\mathbf{y}_t | \mathbf{z}_{t,n}^{(\ell)}, \mathbf{w}_t^{(m)}) \right] \right. \right.$$
$$\left. + \mathbb{D}_{\text{KL}} \left[ q_\phi(\mathbf{z}_{t,n} | \mathbf{x}_{t,n}) || \sum_{i \neq t} \beta_{ti} q_\phi(\mathbf{z}_{t,n} | \mathbf{x}_{t,n}, \mathcal{D}_i) \right] \right\} + \mathbb{D}_{\text{KL}} \left[ q_\theta(\mathbf{w}_t | \mathcal{D}_t) || \sum_{i \neq t} \alpha_{ti} q_\theta(\mathbf{w}_i | \mathcal{D}_i) \right] \right\},$$
$$(11)$$

where $\mathbf{z}_{t,n}^{(\ell)} \sim q_\phi(\mathbf{z}_{t,n} | \mathbf{x}_{t,n})$ and $\mathbf{w}_t^{(m)} \sim q_\theta(\mathbf{w}_t | \mathcal{D}_t)$. To sample from the variational posteriors, we adopt the reparameterization trick [28]. $L$ and $M$ are the number of Monte Carlo samples for the variational posteriors of latent representations and classifiers, respectively. In practice, $L$ and $M$ are set to 10, which yields good performance while being computationally efficient. We investigate the sensitivity of $L$ and $M$ in the supplementary materials. $\theta$ represents the statistical parameters associated with the classifiers or the inference parameters for the amortized classifier; $\phi$ denotes parameters of the shared inference network for the latent representation.

We minimize this empirical objective function to optimize the model parameters jointly. The log-likelihood term is implemented as the cross-entropy loss. In the practical implementation of the KL terms, we adopt the closed-form solution based on its upper bound as done in [35], e.g., $\mathbb{D}_{\text{KL}} \left[ q(\mathbf{w}_t) || \sum_{i \neq t} \alpha_{ti} q(\mathbf{w}_i) \right] \leq \sum_{i \neq t} \alpha_{ti} \mathbb{D}_{\text{KL}} \left[ q(\mathbf{w}_t) || q(\mathbf{w}_i) \right]$. Minimizing the KL terms encourages the model to leverage shared knowledge provided from related tasks at the instance level for representations and at the categorical level for classifiers.

At test time, we obtain the prediction for a test sample $\mathbf{x}_t$ by calculating the following probability using Monte Carlo sampling:

$$p(\mathbf{y}_t | \mathbf{x_t}) \approx \frac{1}{ML} \sum_{l=1}^{L} \sum_{m=1}^{M} p(\mathbf{y} | \mathbf{z}_t^{(l)}, \mathbf{w}_t^{(m)}), \tag{12}$$

where we draw samples from posteriors: $\mathbf{z}_t^{(l)} \sim q_\phi(\mathbf{z} | \mathbf{x})$ and $\mathbf{w}_t^{(m)} \sim q_\theta(\mathbf{w} | \mathcal{D}_t)$.

## 3 Related Works

**Multi-Task Learning Settings**  There are mainly two typical multi-task learning settings: single-input multi-output (SIMO) and multi-input multi-output (MIMO). In the first setting different tasks are assumed to share the same input data during both training and testing [52]. Recently, this setting has been integrated into deep neural networks to boost performance for multi-label prediction [38, 42] and various pixel-level prediction tasks, such as depth prediction, semantic segmentation and surface normal prediction [9, 14, 25, 32, 34, 50]. In the multi-input multi-output setting, each task has its own input from a different domain and output [33, 55, 3, 51]. Tasks are related by sharing the same target space, e.g., class label space or regression target space. This setting, which is the focus of our work, is made challenging by the distribution shift between different tasks.

**Probabilistic Multi-Task Learning**  This has been widely developed to explore shared priors for all tasks. The relationships among multiple tasks are investigated by designing priors over model parameters [49, 45, 1, 3] under the Bayesian framework, or sharing the covariance structure of parameters [11]. In particular, Bakker et al. [3] adopted a Bayesian approach which introduces a joint learnable prior distribution that allows similar tasks to share model parameters and others to be more loosely connected. In contrast to these works, our probabilistic modeling explores task relatedness by specifying priors under the variational Bayesian inference framework.

**Regularization-based Multi-Task Learning**  This has been extensively investigated [52], with the aim of exploring different constraints, e.g., $l_{\infty,1}$ and $l_{p,q}$ norms, imposed on model parameters or features as regularizers [2, 30, 37, 24, 23, 21]. In doing so, similar tasks are encouraged to share the same parameter patterns or the same subset of features. In particular, Long et al. [33] exploited

the task relatedness underlying parameter tensors to regularize the maximum a posteriori estimation. This method was also developed for the multi-input multi-output setting. In our work, we explore shared knowledge for feature representations and classifiers/regressors jointly in a single framework.

**Gumbel-Softmax Technique**   This technique [22] was recently introduced into multi-task learning for exploring task relationships. In some multi-task learning models, it is used for searching the optimal architecture for each task by assigning kernels in each convolutional layer [7], choosing the child node during the forward pass [19], or controlling whether the sub-module is skipped [43]. These works were exclusively for the single-input multi-output setting. In our work, we also adopt the Gumbel-Softmax technique for multi-task learning, using it to learn different correlation patterns for individual tasks. In this way, each task is encouraged to find the most relevant tasks for leveraging the shared knowledge, while reducing interference by irrelevant tasks.

# 4   Experiments

We conduct extensive experiments to evaluate the proposed VMTL on five benchmark datasets for both classification and regression tasks. We perform comprehensive comparisons with previous methods and ablation studies to gain insights into the effectiveness of our methods. We report the major experimental results here and provide more results in the supplementary materials.

## 4.1   Datasets

**Office-Home** [47] contains images from four domains/tasks: Artistic (A), Clipart (C), Product (P) and Real-world (R). Each task contains images from 65 object categories collected under office and home settings. There are about $15, 500$ images in total.

**Office-Caltech** [16] contains the ten categories shared between Office-31 [39] and Caltech-256 [18]. One task uses data from Caltech-256 (C), and the other three tasks use data from Office-31, whose images were collected from three distinct domains/tasks, namely Amazon (A), Webcam (W) and DSLR (D). There are $8 \sim 151$ samples per category per task, and $2, 533$ images in total.

**ImageCLEF** [33], the benchmark for the ImageCLEF domain adaptation challenge, contains 12 common categories shared by four public datasets/tasks: Caltech-256 (C), ImageNet ILSVRC 2012 (I), Pascal VOC 2012 (P), and Bing (B). There are $2, 400$ images in total.

**DomainNet** [36] is a large-scale dataset with approximately 0.6 million images distributed among 345 categories. It contains six distinct domains: Clipart (C), Infograph (I), Painting (P), Quickdraw (Q), Real (R) and Sketch (S). Due to the large number of object categories, this is an extremely challenging benchmark for multi-task learning with limited data. To the best of our knowledge, it has therefore not been used for multi-task learning before.

**Rotated MNIST** [29] is adopted for angle regression tasks. Each digit denotes one task and the tasks are related because they share the same rotation angle space. Each image is rotated by $0°$ through $90°$ in intervals of $10°$, where the rotation angle is the regression target.

## 4.2   Experimental Settings

**Implementation Details**   We adopt the standard evaluation protocols [52] for the *Office-Home*, *Office-Caltech* and *ImageCLEF* datasets, randomly selecting $5\%$, $10\%$, and $20\%$ of samples from each task in the dataset as the training set, using the remaining samples as the test set [33]. Note that when we use $5\%$, $10\%$ and $20\%$ of labeled data for training, there are on average about 3, 6 and 12 samples per category per task, respectively. For the large-scale *DomainNet* dataset, we set the splits to $1\%$, $2\%$ and $4\%$, which also results in an average of 3, 6 and 12 samples per category per task, respectively. For the regression dataset, *Rotated MNIST*, we set the splits to $0.1\%$ and $0.2\%$, which results in an average of 6 and 12 samples per task per angle, respectively. For a fair comparison, all methods in our paper share the same split documents.

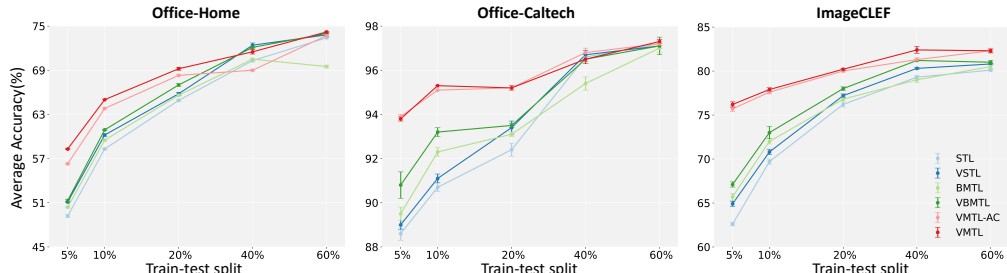

Fig. 2: The performance in terms of average accuracy with a 95% confidence interval under different proportions of training data on *Office-Home*, *Office-Caltech* and *ImageCLEF*. Best viewed in color.

Table 1: Effectiveness of variational Bayesian representations ($\mathbf{z}$) and classifiers ($\mathbf{w}$).

| $\mathbf{z}$ | $\mathbf{w}$ | 5% | 10% | 20% |
|---|---|---|---|---|
| $\times$ | $\times$ | $50.4_{\pm 0.1}$ | $59.5_{\pm 0.1}$ | $65.6_{\pm 0.1}$ |
| $\checkmark$ | $\times$ | $57.4_{\pm 0.0}$ | $64.4_{\pm 0.1}$ | $68.5_{\pm 0.1}$ |
| $\times$ | $\checkmark$ | $56.8_{\pm 0.1}$ | $63.7_{\pm 0.1}$ | $68.5_{\pm 0.1}$ |
| $\checkmark$ | $\checkmark$ | $\mathbf{58.3_{\pm 0.1}}$ | $\mathbf{65.0_{\pm 0.0}}$ | $\mathbf{69.2_{\pm 0.2}}$ |

Table 2: Performance comparison of our Gumbel-Softmax priors with other alternative priors.

| Priors | 5% | 10% | 20% |
|---|---|---|---|
| Mean | $57.5_{\pm 0.0}$ | $64.4_{\pm 0.1}$ | $68.5_{\pm 0.1}$ |
| Learnable weighted | $57.2_{\pm 0.2}$ | $64.2_{\pm 0.2}$ | $68.9_{\pm 0.2}$ |
| Gumbel-Softmax | $\mathbf{58.3_{\pm 0.1}}$ | $\mathbf{65.0_{\pm 0.0}}$ | $\mathbf{69.2_{\pm 0.2}}$ |

Following the experimental settings in [33], we adopt the VGGnet [41], remove the final classifier layer and employ the remaining model to pre-extract the input feature $\mathbf{x}$. From the input, we infer its latent representation $\mathbf{z}$ using amortized inference by multi-layer perceptrons (MLPs) [28]. In our experiments, the temperature of the Gumbel-Softmax priors (6) and (9) is annealed using the same schedule applied in [22]: we start with a high temperature and gradually anneal it to a small but non-zero value. For the KL-divergence in (11), we use the annealing scheme from [6].

We adopt the Adam optimizer [27] with a learning rate of 1e-4 for training. All the results are obtained based on a 95% confidence interval from five runs. The detailed network architectures are given in the supplementary materials. The code will be available at https://github.com/autumn9999/VMTL.git.

**Baselines** Since the multi-task learning setting with limited training data is relatively new, most previous methods designed under other settings are not directly applicable due to the distribution shift between different tasks. To enable a comprehensive comparison, we implement several strong baselines including both Bayesian and non-Bayesian approaches. The single-task learning (STL) is implemented by task-specific feature extractors and classifiers without sharing knowledge among tasks. The basic multi-task learning (BMTL) is a deterministic model, which has a shared feature extractor and task-specific classifiers. Meanwhile, we define variational Bayesian extensions of STL and BMTL, which we name VSTL and VBMTL, by placing priors as uninformative standard Gaussian distributions (details can be found in the supplementary materials). VSTL and VBMTL serve as baselines to directly demonstrate the benefits of the probabilistic modeling of multi-task learning based on variational Bayesian inference. We use VMTL-AC to represent our proposed method with amortized classifiers. We compare against four representative methods [3, 25, 33, 38], which are implemented by following the same experimental setup as our methods. Details of the implementation can be found in supplementary materials.

### 4.3  Results

We demonstrate the effectiveness of our VMTL in handling very limited data, the benefits of variational Bayesian representations and classifiers, and the effectiveness of Gumbel-Softmax priors in learning task relationships. Detailed information can be found in the supplementary materials.

**Effectiveness in Handling Limited Data** Our methods are highly effective at handling scenarios with very limited data. To show this, we compare our VMTL and VMTL-AC with other baselines, including single-task baselines (STL and VSTL) and multi-task baselines (BMTL and VBMTL) on the *Office-Home, Office-Caltech* and *ImageCLEF* datasets. We employ a large range of train-test splits, from 5% to 60%. As shown in Fig. 2, our proposed VMTL outperforms the baselines on all three datasets with limited data (less than 20%) by large margins. The proposed VMTL-AC achieves comparable performance to VMTL on most benchmarks, despite the amortization gap.

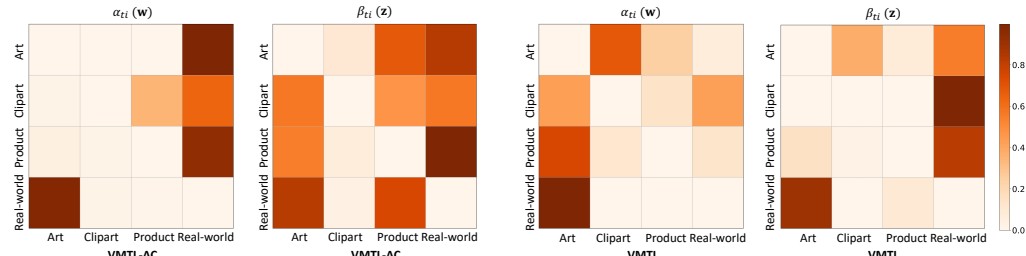

Fig. 3: Tasks show different correlation patterns $\alpha_t$ and $\beta_t$ for classifiers ($\mathbf{w}$) and representations ($\mathbf{z}$). Similar tasks are more correlated, e.g., `Real_World` is closer to `Art` than `Clipart`.

We observe that the proposed methods not only show dominant performance in the setting with limited data but also produce comparable even better performance than baselines in the setting with large amounts of data, which demonstrates the effectiveness of our methods in handling challenging scenarios with limited training data. The proposed methods show less significant improvements under the setting with larger amounts of training data than the setting with limited data. The reason could be that the knowledge transfer becomes less crucial since individual tasks suffer less from over-fitting caused by limited data. Compared with other multi-task learning methods, the proposed methods still show certain improvements under the setting with large amounts of training data.

**Effectiveness of Variational Bayesian Approximation** We investigate the effect of variational Bayesian inference for representations and classifiers separately. We conduct these experiments on the *Office-Home* dataset. The results with different train-test splits are shown in Table 1, which indicates whether $\mathbf{z}$ and $\mathbf{w}$ are probabilistic or not, e.g., a $\times$ for $\mathbf{w}$ means a deterministic classifier. More detailed experimental results are provided in the supplementary materials. As can be seen, both variational Bayesian representations and classifiers can benefit performance. This benefit becomes more significant when training data is very limited, which indicates the effectiveness of leveraging shared knowledge by conditioning priors on related tasks. In addition, in Fig. 2, VSTL demonstrates to be a very strong Bayesian model, which again demonstrates the benefits of variational Bayesian representations and classifiers, even without explicitly leveraging shared knowledge among tasks.

**Effectiveness of Gumbel-Softmax Priors** The introduced Gumbel-Softmax priors provide an effective way to learn data-driven task relationships. To demonstrate this, we compare with two alternatives without the Gumbel-Softmax technique in Table 2, including the mean and the learnable weighted posteriors of other tasks. The proposed Gumbel-Softmax priors perform the best, consistently surpassing other alternatives. We would like to highlight that the advantage of Gumbel-Softmax priors is even larger under very limited training data, e.g., $5\%$, which creates a challenging scenario where it is crucial to leverage shared knowledge provided by related tasks. To further investigate the learned Gumbel-Softmax priors, we visualize the values of the mixing weights $\alpha$ and $\beta$ for both VMTL and VMTL-AC. The results on *Office-Home* with the $5\%$ split are shown in Fig. 3. We can see that the relatedness among tasks is relatively sparse, which could avoid interference by irrelevant tasks. It is worth noting that the mixing weight between tasks is higher when they are more correlated. For instance, the task *Real_World* is closer to *Art* than to *Clipart*.

**Comparison with Other Methods** The comparison results on the small-scale *Office-Home*, *Office-Caltech*, *ImageCLEF* datasets and large-scale *DomainNet* dataset are shown in Tables 3, 4, 5 and 6, respectively. The average accuracy of all tasks is used for overall performance measurement. The best results are marked in bold, while the second-best are marked by underlining.

Our proposed methods achieve the best performance on all four datasets with different train-test splits. On the most challenging setting with $5\%$ training data, VMTL shows a large performance advantage over the baselines. This demonstrates the effectiveness of VMTL in exploring relatedness to improve the performance of each task from limited data. VMTL-AC can also produce comparable performance and is better than most previous methods.

We would like to highlight that Bakker et al. [3], a Bayesian method for multi-task learning optimized by an expectation-maximization algorithm, produce very competitive performance. Long et al. [33] use point estimation to find approximate values of model parameters. However, our methods

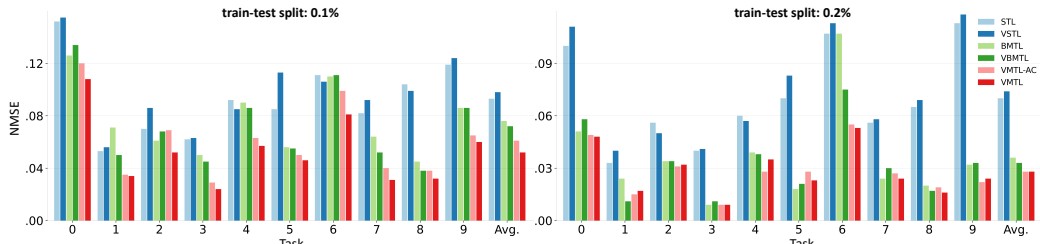

Fig. 4: Performance comparison (normalized mean squared errors) for rotation angle regression.

Table 3: Performance comparison of different methods on the *Office-Home* dataset.

| Methods | 5% | | | | | 10% | | | | | 20% | | | | |
|---|---|---|---|---|---|---|---|---|---|---|---|---|---|---|---|
| | A | C | P | R | Avg. | A | C | P | R | Avg. | A | C | P | R | Avg. |
| STL | 36.7±0.4 | 30.8±0.5 | 67.5±0.3 | 61.7±0.3 | 49.2±0.2 | 50.4±0.3 | 40.8±0.3 | 74.4±0.4 | 67.5±0.4 | 58.3±0.1 | 54.6±0.4 | 50.6±0.4 | 81.3±0.2 | 73.1±0.3 | 64.9±0.1 |
| VSTL | 37.9±0.3 | 33.3±0.5 | 69.0±0.6 | 64.0±0.2 | 51.1±0.1 | 52.0±0.4 | 43.1±0.2 | 76.2±0.4 | 69.4±0.3 | 60.2±0.2 | 55.9±0.3 | 52.0±0.5 | 81.2±0.5 | 73.8±0.2 | 65.8±0.2 |
| Bakker et al. [3] | 40.0±0.1 | 33.6±0.4 | 69.8±0.4 | 63.6±0.3 | 52.8±0.1 | 52.5±0.3 | 42.3±0.4 | 75.7±0.6 | 69.5±0.5 | 60.0±0.2 | 61.3±0.2 | 56.5±0.2 | 81.7±0.3 | 75.4±0.2 | 68.7±0.2 |
| Long et al. [33] | 47.8±0.4 | 37.9±0.2 | 73.6±0.3 | 70.4±0.2 | 57.4±0.1 | 57.2±0.3 | 43.3±0.1 | 78.7±0.3 | 74.4±0.1 | 63.4±0.2 | 65.1±0.3 | 46.7±0.2 | 79.9±0.3 | 76.6±0.3 | 67.1±0.1 |
| Kendall et al. [25] | 40.2±0.2 | 33.6±0.4 | 69.5±0.2 | 63.7±0.1 | 51.8±0.1 | 49.1±0.1 | 38.7±0.3 | 73.4±0.2 | 67.4±0.3 | 57.2±0.2 | 59.5±0.3 | 53.8±0.3 | 80.1±0.1 | 73.6±0.4 | 66.8±0.2 |
| Qian et al. [38] | 37.9±0.3 | 31.4±0.2 | 67.7±0.3 | 62.4±0.2 | 49.9±0.2 | 47.1±0.2 | 37.2±0.1 | 70.5±0.2 | 66.3±0.3 | 55.3±0.1 | 58.3±0.2 | 53.5±0.3 | 79.8±0.2 | 73.1±0.3 | 66.2±0.1 |
| BMTL | 37.6±0.4 | 31.5±0.3 | 68.5±0.2 | 63.8±0.2 | 50.4±0.1 | 51.0±0.2 | 41.6±0.1 | 76.0±0.3 | 69.2±0.3 | 59.5±0.1 | 56.6±0.3 | 51.8±0.5 | 80.9±0.3 | 72.9±0.4 | 65.6±0.1 |
| VBMTL | 38.6±0.5 | 32.8±0.8 | 69.2±0.1 | 64.5±0.3 | 51.3±0.1 | 53.2±0.2 | 43.1±0.5 | 76.6±0.3 | 70.5±0.5 | 60.9±0.1 | 58.2±0.3 | 53.1±0.4 | 81.9±0.2 | 74.8±0.3 | 67.0±0.2 |
| **VMTL-AC** | 51.6±0.4 | 37.6±0.2 | 69.8±0.3 | 66.5±0.1 | 56.3±0.1 | 58.3±0.4 | 47.0±0.4 | 77.2±0.3 | 72.8±0.3 | 63.8±0.1 | 62.2±0.2 | 53.3±0.3 | 81.9±0.2 | 75.7±0.1 | 68.3±0.1 |
| **VMTL** | 53.5±0.1 | 39.2±0.2 | 71.5±0.3 | 69.0±0.1 | **58.3±0.1** | 60.2±0.2 | 47.5±0.2 | 78.2±0.3 | 74.0±0.2 | **65.0±0.0** | 64.0±0.3 | 53.5±0.4 | 82.5±0.3 | 76.8±0.1 | **69.2±0.2** |

Table 4: Performance comparison of different methods on the *Office-Caltech* dataset.

| Methods | 5% | | | | | 10% | | | | | 20% | | | | |
|---|---|---|---|---|---|---|---|---|---|---|---|---|---|---|---|
| | A | W | D | C | Avg. | A | W | D | C | Avg. | A | W | D | C | Avg. |
| STL | 87.4±0.4 | 87.9±0.3 | 96.4±0.5 | 82.8±0.6 | 88.6±0.3 | 92.8±0.5 | 97.7±0.3 | 87.8±0.2 | 84.3±0.4 | 90.7±0.2 | 94.9±0.2 | 92.8±0.6 | 95.2±0.6 | 86.7±0.6 | 92.4±0.3 |
| VSTL | 88.3±0.3 | 89.1±0.4 | 97.0±0.2 | 81.4±0.5 | 89.0±0.2 | 93.1±0.2 | 96.6±0.4 | 90.0±0.5 | 84.5±0.3 | 91.1±0.2 | 95.5±0.4 | 94.5±0.2 | 96.0±0.6 | 87.7±0.4 | 93.4±0.3 |
| Bakker et al. [3] | 93.2±0.2 | 94.0±0.3 | 94.7±0.3 | 85.4±0.4 | 91.8±0.1 | 94.9±0.4 | 97.6±0.5 | 96.6±0.5 | 90.9±0.4 | 95.0±0.2 | 95.2±0.2 | 94.4±0.4 | 99.5±0.3 | 91.3±0.1 | 95.1±0.1 |
| Long et al. [33] | 92.7±0.2 | 94.3±0.2 | 97.1±0.2 | 89.2±0.6 | 93.4±0.2 | 95.0±0.3 | 98.1±0.4 | 95.0±0.5 | 91.3±0.2 | 94.8±0.3 | 95.5±0.3 | 94.9±0.1 | 99.2±0.3 | 91.0±0.4 | 95.1±0.1 |
| Kendall et al. [25] | 93.6±0.4 | 92.5±0.2 | 95.0±0.5 | 83.9±0.5 | 91.2±0.3 | 94.9±0.6 | 96.2±0.4 | 93.6±0.3 | 90.4±0.2 | 93.8±0.2 | 95.4±0.7 | 93.2±0.4 | 99.2±0.4 | 91.2±0.3 | 94.7±0.3 |
| Qian et al. [38] | 92.6±0.3 | 90.9±0.2 | 95.7±0.4 | 85.2±0.6 | 91.1±0.2 | 94.2±0.4 | 97.0±0.4 | 95.0±0.3 | 90.2±0.3 | 94.1±0.3 | 95.7±0.4 | 94.1±0.2 | 99.2±0.5 | 91.1±0.4 | 95.0±0.2 |
| BMTL | 90.0±0.7 | 89.4±0.8 | 95.0±1.1 | 83.5±0.5 | 89.5±0.3 | 93.6±0.1 | 97.0±0.6 | 92.1±0.7 | 86.3±0.4 | 92.3±0.2 | 95.0±0.2 | 94.5±0.7 | 96.0±1.2 | 86.8±0.2 | 93.1±0.1 |
| VBMTL | 90.3±1.5 | 91.3±0.5 | 97.1±0.0 | 84.4±0.9 | 90.8±0.6 | 94.2±0.1 | 97.0±0.0 | 93.8±0.5 | 87.9±0.3 | 93.2±0.2 | 95.5±0.1 | 94.6±0.3 | 96.0±0.9 | 87.8±0.4 | 93.5±0.1 |
| **VMTL-AC** | 93.5±0.4 | 95.4±0.3 | 96.7±0.3 | 90.0±0.3 | **93.9±0.1** | 94.9±0.1 | 97.1±0.4 | 97.9±0.0 | 90.6±0.3 | 95.1±0.0 | 95.5±0.2 | 96.4±0.4 | 98.6±0.3 | 90.3±0.4 | **95.2±0.1** |
| **VMTL** | 93.7±0.2 | 95.2±0.3 | 96.4±0.4 | 89.7±0.4 | 93.8±0.1 | 95.4±0.1 | 97.6±0.1 | 97.4±0.4 | 90.9±0.2 | **95.3±0.0** | 95.6±0.2 | 95.6±0.4 | 98.4±0.5 | 91.1±0.3 | 95.2±0.1 |

Table 5: Performance comparison of different methods on the *ImageCLEF* dataset.

| Methods | 5% | | | | | 10% | | | | | 20% | | | | |
|---|---|---|---|---|---|---|---|---|---|---|---|---|---|---|---|
| | C | I | P | B | Avg. | C | I | P | B | Avg. | C | I | P | B | Avg. |
| STL | 85.4±0.6 | 71.4±0.4 | 57.7±0.2 | 36.0±0.2 | 62.6±0.2 | 88.9±0.5 | 77.8±0.3 | 64.3±0.2 | 47.6±0.5 | 69.7±0.3 | 92.9±0.6 | 84.6±0.3 | 72.5±0.4 | 54.6±0.6 | 76.2±0.3 |
| VSTL | 87.0±0.4 | 73.2±0.5 | 60.5±0.4 | 39.0±0.5 | 64.9±0.3 | 89.6±0.3 | 79.1±0.6 | 66.6±0.3 | 48.0±0.2 | 70.8±0.3 | 93.3±0.3 | 87.3±0.3 | 72.7±0.5 | 55.4±0.5 | 77.2±0.2 |
| Bakker et al. [3] | 90.9±0.4 | 85.4±0.6 | 68.1±0.3 | 51.4±0.5 | 73.9±0.3 | 91.0±0.5 | 87.1±0.3 | 73.4±0.4 | 54.5±0.2 | 76.5±0.4 | 94.4±0.5 | 90.6±0.4 | 74.2±0.4 | 57.9±0.3 | 79.3±0.4 |
| Long et al. [33] | 90.1±0.5 | 76.5±0.5 | 72.8±0.3 | 54.9±0.4 | 73.7±0.4 | 93.3±0.4 | 83.2±0.6 | 70.4±0.4 | 56.3±0.4 | 75.8±0.2 | 94.4±0.4 | 89.2±0.5 | 75.8±0.5 | 59.4±0.3 | 79.7±0.3 |
| Kendall et al. [25] | 93.2±0.6 | 86.1±0.4 | 68.6±0.3 | 50.4±0.4 | 74.6±0.2 | 91.9±0.3 | 88.9±0.6 | 74.3±0.3 | 52.4±0.2 | 76.9±0.3 | 93.3±0.4 | 91.0±0.2 | 75.6±0.2 | 56.9±0.4 | 79.2±0.3 |
| Qian et al. [38] | 91.6±0.3 | 85.8±0.4 | 68.4±0.3 | 50.2±0.4 | 74.0±0.4 | 90.7±0.4 | 88.1±0.6 | 75.6±0.4 | 54.6±0.3 | 77.3±0.3 | 93.1±0.3 | 92.1±0.5 | 74.4±0.7 | 55.8±0.6 | 78.9±0.5 |
| BMTL | 88.3±0.5 | 71.2±0.9 | 61.2±0.9 | 40.0±0.8 | 65.7±0.4 | 90.6±0.8 | 79.3±0.2 | 66.3±0.5 | 51.9±0.6 | 72.0±0.3 | 95.0±0.2 | 86.5±0.2 | 71.9±0.8 | 56.0±0.6 | 76.8±0.3 |
| VBMTL | 88.8±0.7 | 75.0±0.8 | 62.9±0.3 | 41.7±1.6 | 67.1±0.3 | 91.1±0.2 | 80.7±0.6 | 69.0±0.8 | 51.1±1.3 | 73.0±0.7 | 93.7±0.3 | 87.9±0.6 | 73.7±1.2 | 56.7±0.2 | 78.0±0.2 |
| **VMTL-AC** | 89.3±0.6 | 81.5±0.8 | 72.3±0.4 | 59.5±0.8 | 75.7±0.3 | 92.8±0.4 | 85.9±0.3 | 71.9±0.6 | 59.9±0.3 | 77.6±0.2 | 92.9±0.3 | 88.0±0.4 | 78.6±0.2 | 60.5±0.2 | 80.0±0.1 |
| **VMTL** | 91.5±0.2 | 83.5±0.6 | 71.6±0.8 | 58.2±0.7 | **76.2±0.3** | 94.2±0.2 | 86.0±0.5 | 71.7±0.6 | 59.8±0.4 | **77.9±0.2** | 93.9±0.1 | 89.4±0.2 | 78.0±0.4 | 59.5±0.4 | **80.2±0.1** |

Table 6: Performance comparison of different methods on the large-scaled *DomainNet* dataset.

| Methods | 1% | | | | | | | 2% | | | | | | |
|---|---|---|---|---|---|---|---|---|---|---|---|---|---|---|
| | C | I | P | Q | R | S | Avg. | C | I | P | Q | R | S | Avg. |
| STL | 15.0±0.3 | 4.0±0.4 | 19.7±0.3 | 7.5±0.2 | 50.6±0.2 | 9.6±0.4 | 17.7±0.2 | 20.5±0.2 | 6.6±0.3 | 26.0±0.4 | 7.2±0.2 | 56.5±0.3 | 14.0±0.3 | 21.8±0.3 |
| VSTL | 18.9±0.2 | 5.4±0.3 | 23.5±0.2 | 15.2±0.4 | 54.5±0.3 | 12.2±0.3 | 21.6±0.1 | 26.4±0.1 | 8.9±0.3 | 30.9±0.3 | 20.2±0.3 | 60.7±0.2 | 17.8±0.1 | 27.5±0.1 |
| Bakker et al. [3] | 20.2±0.2 | 5.8±0.1 | 26.2±0.2 | 21.0±0.2 | 52.8±0.2 | 14.7±0.1 | 23.5±0.1 | 23.4±0.1 | 8.6±0.1 | 32.5±0.3 | 21.5±0.3 | 58.7±0.2 | 19.2±0.2 | 27.3±0.1 |
| Long et al. [33] | 19.1±0.2 | 8.1±0.1 | 30.7±0.2 | 5.7±0.1 | 57.1±0.2 | 15.1±0.1 | 22.6±0.1 | 24.3±0.3 | 11.0±0.2 | 36.6±0.1 | 7.0±0.2 | 60.9±0.1 | 18.0±0.3 | 26.3±0.2 |
| Kendall et al. [25] | 17.5±0.1 | 4.5±0.1 | 24.4±0.2 | 19.4±0.3 | 50.7±0.1 | 11.6±0.1 | 21.3±0.1 | 23.5±0.2 | 7.7±0.3 | 32.7±0.2 | 22.5±0.2 | 60.2±0.1 | 18.5±0.2 | 27.5±0.1 |
| Qian et al. [38] | 20.6±0.1 | 6.4±0.3 | 27.2±0.2 | 19.6±0.1 | 52.7±0.3 | 13.5±0.2 | 23.3±0.2 | 26.0±0.2 | 8.9±0.2 | 34.5±0.1 | 20.9±0.3 | 62.6±0.2 | 21.0±0.1 | 29.0±0.1 |
| BMTL | 18.3±0.1 | 5.2±0.2 | 22.3±0.1 | 15.0±0.1 | 53.3±0.1 | 11.8±0.2 | 21.0±0.1 | 26.2±0.2 | 8.7±0.2 | 30.3±0.2 | 20.7±0.1 | 59.6±0.2 | 17.8±0.2 | 27.2±0.1 |
| VBMTL | 18.1±0.3 | 5.2±0.1 | 23.0±0.4 | 13.4±0.2 | 53.9±0.3 | 12.0±0.4 | 20.9±0.3 | 26.2±0.5 | 8.9±0.3 | 30.7±0.3 | 17.5±0.2 | 59.9±0.4 | 17.5±0.1 | 26.8±0.2 |
| **VMTL-AC** | 19.8±0.2 | 6.6±0.2 | 24.1±0.2 | 12.1±0.2 | 52.3±0.1 | 13.6±0.1 | 21.4±0.1 | 26.5±0.2 | 9.7±0.2 | 30.6±0.2 | 14.6±0.1 | 58.2±0.2 | 18.3±0.2 | 26.3±0.0 |
| **VMTL** | 25.2±0.1 | 8.8±0.1 | 30.1±0.1 | 13.2±0.3 | 56.4±0.2 | 17.5±0.3 | **25.2±0.1** | 31.1±0.1 | 11.9±0.1 | 35.1±0.1 | 17.1±0.1 | 61.6±0.0 | 21.9±0.1 | **29.8±0.0** |

consistently outperform Bakker et al. [3] and Long et al. [33], which shows the great effectiveness of our variational Bayesian inference for multi-task learning. In addition, the methods [25] and [38] weigh multiple loss functions by considering the homoscedastic uncertainty of each task, which shows the benefit of modeling uncertainty, while they are also outperformed by our methods.

It is also worth mentioning that VMTL-AC demonstrates computational advantages as well, with faster convergence than VMTL due to amortized learning. Besides, we find that VMTL-AC demonstrates good robustness against adversarial attacks. This could be due to the fact that amortized learning applies the mean feature representations to generate classifiers, which is more robust to attacks. Due to space limit, more experiments and detailed discussions about the amortized classifier are provided in the supplementary materials.

In addition, we apply our methods to a regression task on *Rotated MNIST* for rotation angle estimation. The comparison results are shown in Fig. 4. The proposed VMTL and VMTL-AC outperform other methods with lower average NMSE. For some digits/tasks, e.g. $0, 6, 9$, their NMSE are higher than other digits/tasks. The reason could be that these digits have some degree of rotational invariance, which makes it harder for them to regress their target angles.

## 5 Conclusion

In this paper, we develop variational multi-task learning (VMTL), a novel variational Bayesian inference approach for simultaneously learning multiple tasks. We propose a probabilistic latent variable model to explore task relatedness, where we introduce stochastic variables as the task-specific classifiers and latent representations in a single Bayesian framework. In doing so, multi-task learning is cast as a variational inference problem, which provides a unified way to jointly explore knowledge shared among representations and classifiers by specifying priors. We further introduce Gumbel-Softmax priors, which offer an effective way to learn the task relatedness in a data-driven manner for each task. We evaluate VMTL on five benchmark datasets of multi-task learning under both classification and regression. Results demonstrate the effectiveness of our variational multi-task learning and our method delivers state-of-the-art performance.

## Acknowledgment

This work is financially supported by the Inception Institute of Artificial Intelligence, the University of Amsterdam and the allowance Top consortia for Knowledge and Innovation (TKIs) from the Netherlands Ministry of Economic Affairs and Climate Policy.

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
