# Supplementary Materials for "Variational Multi-Task Learning with Gumbel-Softmax Priors"

**Jiayi Shen**[1], **Xiantong Zhen**[1,2], **Marcel Worring**[1], **Ling Shao**[2]
[1]AIM Lab, University of Amsterdam, Netherlands
[2]Inception Institute of Artificial Intelligence, Abu Dhabi, UAE

## A Derivation

### A.1 Evidence Lower Bound for Multi-Task Learning

In this paper, we follow the multi-input multi-output data setting [33, 52] for multi-task learning, each task $t$ has its own training data $\mathcal{D}_t = \{\mathbf{x}_{t,n}, \mathbf{y}_{t,n}\}_{n=1}^{N_t}$. Note that we derive our methodology mainly using terminologies related to classification tasks, but it is also applicable to regression tasks.

Under the probabilistic formulation for multi-task learning, we start with the conditional log-likelihood for each task $t$: $\log p(\mathbf{y}_{t,n}|\mathbf{x}_{t,n}, \mathcal{D}_{1:T\backslash t})$, where $(\mathbf{x}_{t,n}, \mathbf{y}_{t,n})$ is a sample from the data of current task $t$ and $\mathcal{D}_{1:T\backslash t}$ is the data from all related tasks.

Here, we introduce the latent variables $\mathbf{z}_{t,n}$ and $\mathbf{w}_t$:

$$\log p(\mathbf{y}_{t,n}|\mathbf{x}_{t,n}, \mathcal{D}_{1:T\backslash t}) = \log \int \int p(\mathbf{y}_{t,n}, \mathbf{z}_{t,n}, \mathbf{w}_t, |\mathbf{x}_{t,n}, \mathcal{D}_{1:T\backslash t})d\mathbf{z}_{t,n}d\mathbf{w}_t, \tag{1}$$

where $p(\mathbf{y}_{t,n}, \mathbf{z}_{t,n}, \mathbf{w}_t, |\mathbf{x}_{t,n}, \mathcal{D}_{1:T\backslash t})$ is the joint conditional predictive distribution over the classification label or regression target. Under the assumption that $\mathbf{w}_t$ and $\mathbf{z}_{t,n}$ are conditionally independent, we obtain

$$\log p(\mathbf{y}_{t,n}|\mathbf{x}_{t,n}, \mathcal{D}_{1:T\backslash t}) = \log \int \int p(\mathbf{y}_{t,n}|\mathbf{z}_{t,n}, \mathbf{w}_t)p(\mathbf{z}_{t,n}|\mathbf{x}_{t,n}, \mathcal{D}_{1:T\backslash t})p(\mathbf{w}_t|\mathcal{D}_{1:T\backslash t})d\mathbf{z}_{t,n}d\mathbf{w}_t. \tag{2}$$

Next, we introduce the variational posteriors $q_\phi(\mathbf{z}_{t,n}|\mathbf{x}_{t,n})$ and $q_\theta(\mathbf{w}_t|\mathcal{D}_t)$ to approximate the true posteriors for latent representations and classifiers, respectively. By leveraging Jensen's inequality, we have the following steps as

$$
\begin{aligned}
&\log p(\mathbf{y}_{t,n}|\mathbf{x}_{t,n}, \mathcal{D}_{1:T\backslash t}) \\
&= \log \int \int p(\mathbf{y}_{t,n}|\mathbf{z}_{t,n}, \mathbf{w}_t)p(\mathbf{z}_{t,n}|\mathbf{x}_{t,n}, \mathcal{D}_{1:T\backslash t})d\mathbf{z}_{t,n}p(\mathbf{w}_t|\mathcal{D}_{1:T\backslash t})\frac{q_\theta(\mathbf{w}_t|\mathcal{D}_t)}{q_\theta(\mathbf{w}_t|\mathcal{D}_t)}d\mathbf{w}_t \\
&\geq \int \log \Big[ \frac{\int p(\mathbf{y}_{t,n}|\mathbf{z}_{t,n}, \mathbf{w}_t)p(\mathbf{z}_{t,n}|\mathbf{x}_{t,n}, \mathcal{D}_{1:T\backslash t})d\mathbf{z}_{t,n}p(\mathbf{w}_t|\mathcal{D}_{1:T\backslash t})}{q_\theta(\mathbf{w}_t|\mathcal{D}_t)} \Big] q_\theta(\mathbf{w}_t|\mathcal{D}_t)d\mathbf{w}_t \\
&= \mathbb{E}_{q_\theta}\Big[ \log \int p(\mathbf{y}_{t,n}|\mathbf{z}_{t,n}, \mathbf{w}_t)p(\mathbf{z}_{t,n}|\mathbf{x}_{t,n}, \mathcal{D}_{1:T\backslash t})\frac{q_\phi(\mathbf{z}_{t,n}|\mathbf{x}_{t,n})}{q_\phi(\mathbf{z}_{t,n}|\mathbf{x}_{t,n})}d\mathbf{z}_{t,n}\Big] \\
&\quad - \mathbb{KL}[q_\theta(\mathbf{w}_t|\mathcal{D}_t)||p(\mathbf{w}_t|\mathcal{D}_{1:T\backslash t})] \\
&\geq \mathbb{E}_{q_\theta}\mathbb{E}_{q_\phi}[\log p(\mathbf{y}_{t,n}|\mathbf{z}_{t,n}, \mathbf{w}_t)] - \mathbb{KL}[q_\phi(\mathbf{z}_{t,n}|\mathbf{x}_{t,n})||p(\mathbf{z}_{t,n}|\mathbf{x}_{t,n}, \mathcal{D}_{1:T\backslash t})] \\
&\quad - \mathbb{KL}[q_\theta(\mathbf{w}_t|\mathcal{D}_t)||p(\mathbf{w}_t|\mathcal{D}_{1:T\backslash t})].
\end{aligned}
\tag{3}
$$

35th Conference on Neural Information Processing Systems (NeurIPS 2021).

Thus, we obtain the ELBO for multi-task learning with latent representations and classifiers as follows:

$$\frac{1}{T}\sum_{t=1}^{T}\log p(\mathcal{Y}_t|\mathcal{X}_t, \mathcal{D}_{1:T\setminus t}) \geq \frac{1}{T}\sum_{t=1}^{T}\Bigg\{\sum_{n=1}^{N_t}\Big\{\mathbb{E}_{q_\theta}\mathbb{E}_{q_\phi}[\log p(\mathbf{y}_{t,n}|\mathbf{z}_{t,n}, \mathbf{w}_t)]$$
$$- \mathbb{KL}[q_\phi(\mathbf{z}_{t,n}|\mathbf{x}_{t,n})||p(\mathbf{z}_{t,n}|\mathbf{x}_{t,n}, \mathcal{D}_{1:T\setminus t})]\Big\} - \mathbb{KL}[q_\theta(\mathbf{w}_t|\mathcal{D}_t)||p(\mathbf{w}_t|\mathcal{D}_{1:T\setminus t})]\Bigg\}. \tag{4}$$

We integrate this ELBO with the proposed Gumbel-Softmax priors to obtain the empirical objective for variational multi-task learning. To verify the effectiveness of our proposed models, we define the basic variational Bayesian multi-task learning (VBMTL) as a baseline. VBMTL shares the inference network of latent representations among tasks but just applies the normal Gaussian as priors of the latent variables.

In this paper, we propose variational multi-task learning, a general probabilistic inference framework, in which we cast multi-task learning as a variational Bayesian inference problem. This general framework can be seamlessly combined with the advantages of other deterministic approaches in leveraging shared knowledge among tasks. We can in fact take advantage of deterministic approaches to generalize even better in more settings, e.g., large amounts of training data. In this case, we can train a large convolutional neural network fully end-to-end to extract more representative features specifically for individual tasks.

### A.2 Evidence Lower Bound for Single-Task Learning

Generally, the proposed Bayesian inference framework which infers the posteriors of presentations $\mathbf{z}$ and classifiers $\mathbf{w}$ jointly can be widely applied in other research fields. For example, based on the proposed Bayesian inference framework, we introduce a variational version of single-task learning (VSTL) and provide the derivation of its evidence lower bound. It is worth noting that single-task learning does not share knowledge among tasks, thus both inference networks of latent representations and classifiers are task-specific. And the log-likelihood for single-task learning is not allowed to be conditioned on the data from other tasks.

$$\log p(\mathbf{y}_{t,n}|\mathbf{x}_{t,n})$$
$$= \log \int\int p(\mathbf{y}_{t,n}|\mathbf{z}_{t,n}, \mathbf{w}_t)p(\mathbf{z}_{t,n}|\mathbf{x}_{t,n})p(\mathbf{w}_t)d\mathbf{z}_{t,n}d\mathbf{w}_t$$
$$= \log \int\int p(\mathbf{y}_{t,n}|\mathbf{z}_{t,n}, \mathbf{w}_t)p(\mathbf{z}_{t,n}|\mathbf{x}_{t,n})d\mathbf{z}_{t,n}p(\mathbf{w}_t)\frac{q_\theta(\mathbf{w}_t)}{q_\theta(\mathbf{w}_t)}d\mathbf{w}_t$$
$$\geq \int \log\Big[\frac{\int p(\mathbf{y}_{t,n}|\mathbf{z}_{t,n}, \mathbf{w}_t)p(\mathbf{z}_{t,n}|\mathbf{x}_{t,n})d\mathbf{z}_{t,n}p(\mathbf{w}_t)}{q_\theta(\mathbf{w}_t)}\Big]q_\theta(\mathbf{w}_t)d\mathbf{w}_t \tag{5}$$
$$= \mathbb{E}_{q_\theta}\Big[\log \int p(\mathbf{y}_{t,n}|\mathbf{z}_{t,n}, \mathbf{w}_t)p(\mathbf{z}_{t,n}|\mathbf{x}_{t,n})\frac{q_\phi(\mathbf{z}_{t,n}|\mathbf{x}_{t,n})}{q_\phi(\mathbf{z}_{t,n}|\mathbf{x}_{t,n})}d\mathbf{z}_{t,n}\Big] - \mathbb{KL}[q_\theta(\mathbf{w}_t)||p(\mathbf{w}_t)]$$
$$\geq \mathbb{E}_{q_\theta}\mathbb{E}_{q_\phi}[\log p(\mathbf{y}_{t,n}|\mathbf{z}_{t,n}, \mathbf{w}_t)] - \mathbb{KL}[q_\phi(\mathbf{z}_{t,n}|\mathbf{x}_{t,n})||p(\mathbf{z}_{t,n}|\mathbf{x}_{t,n})] - \mathbb{KL}[q_\theta(\mathbf{w}_t)||p(\mathbf{w}_t)].$$

In this case, tasks are learned independently with no access to shared knowledge provided by other tasks, thus the priors $p(\mathbf{w}_t)$ and $p(\mathbf{z}_{t,n}|\mathbf{x}_{t,n})$ are set to normal Gaussians as applied in [? ? ? ].

## B More Experimental Details

We train all models and parameters by the Adam optimizer [27] using an NVIDIA Tesla V100 GPU. The learning rate is initially set as $1e-4$ and decreases with a factor of $0.5$ every $3K$ iterations. Details of iteration numbers and batch sizes for different benchmarks are provided in Table B.1. In each batch, the number of training samples from each task and category is identical. The network architectures of our methods for the four benchmarks are given. The code will be available at https://github.com/autumn9999/VMTL.git.

Table B.1. The iteration numbers and batch sizes on different datasets, where $C$ and $T$ denotes the number of classes and tasks in the specific dataset, respectively.

| Dataset | Iteration | Batch size |
|---|---|---|
| *Office-Home* | $15,000$ | $4 * C * T$ |
| *Office-Caltech* | $15,000$ | $4 * C * T$ |
| *ImageCLEF* | $15,000$ | $4 * C * T$ |
| *DomainNet* | $30,000$ | $2 * C * T$ |

Table B.2. The inference network $\theta(\cdot)$ for amortized classifiers in VMTL-AC.

| Output size | Layers |
|---|---|
| 4096 | Input feature |
| 4096 | Dropout (p=0.7) |
| 512 | Fully connected, ELU |
| 512 | Fully connected, ELU |
| 512 | Local reparameterization to $\mu_w, \sigma_w^2$ |

Table B.3. The inference network $\phi(\cdot)$ for latent representations.

| Output size | Layers |
|---|---|
| $4096, N * 4096$ | Input features |
| 4096 | Cross attention |
| 4096 | Dropout (p=0.7) |
| 512 | Fully connected, ELU |
| 512 | Fully connected, ELU |
| 512 | Local reparameterization to $\mu_z, \sigma_z^2$ |

## B.1 Inference Networks

The architecture of the inference network for amortized classifiers in VMTL-AC is in Table B.2. In VMTL, we directly learn the parameters of the distribution of variational posteriors, which has the same dimension as the latent representation. The architecture of the inference network for latent representations is in Table B.3. During inference, we apply the reparameterization trick to generate the samples for both latent variables [**?** ].

## B.2 Implementation of the Compared Previous Works

In this paper, we compare against four representative methods [3, 25, 33, 38], which are implemented by following the same experimental setup as our methods. In practice, we implement the method, Long et al. [33], by applying its open code repository (https://github.com/thuml/MTlearn) under the same experimental environments as ours. For other compared methods, the models consist a shared feature extractor and task-specific classifiers. In particular, Bakker et al. [3] is a Bayesian method for multi-task learning optimized by an expectation-maximization algorithm, producing very competitive performance. Kendall et al. [25] propose to weigh multiple loss functions by considering the homoscedastic uncertainty of each task, which shows the benefit of modeling uncertainty. Qian et al. [38] integrate the variational information bottleneck [**?** ] to the method based on [25], which shows the benefit of exploring shared information for latent representations.

## C More Experimental Results

### C.1 Effectiveness in Handling Limited Data

The results of average accuracy on the *Office-Home*, *Office-Caltech*, and *ImageCLEF* datasets are given in Tables C.4, C.5, and C.6, respectively, which provide detailed information for Fig. 2 of the paper. Our proposed probabilistic models, i.e., VMTL and VMTL-AC outperform the deterministic baseline multi-task learning model (BMTL), which demonstrates the benefits of our proposed

Table C.4. Performance under different proportions of training data on *Office-Home*.

| Methods | 5% | 10% | 20% | 40% | 60% |
|---|---|---|---|---|---|
| STL | 49.2±0.2 | 58.3±0.1 | 64.9±0.1 | 70.3±0.2 | 73.4±0.1 |
| VSTL | 51.1±0.1 | 60.2±0.2 | 65.8±0.2 | **72.4±0.3** | 73.8±0.2 |
| BMTL | 50.4±0.1 | 59.5±0.1 | 65.6±0.1 | 70.5±0.1 | 69.5±0.2 |
| VBMTL | 51.3±0.1 | 60.9±0.1 | 67.0±0.2 | 72.1±0.1 | 74.0±0.1 |
| VMTL-AC | 56.3±0.1 | 63.8±0.1 | 68.3±0.1 | 69.0±0.1 | 73.7±0.2 |
| VMTL | **58.3±0.1** | **65.0±0.0** | **69.2±0.2** | 71.5±0.3 | **74.2±0.1** |

Table C.5. Performance under different proportions of training data on *Office-Caltech*.

| Methods | 5% | 10% | 20% | 40% | 60% |
|---|---|---|---|---|---|
| STL | 88.6±0.3 | 90.7±0.2 | 92.4±0.3 | 96.6±0.2 | 97.2±0.2 |
| VSTL | 89.0±0.2 | 91.1±0.2 | 93.4±0.3 | 96.7±0.2 | 97.1±0.3 |
| BMTL | 89.5±0.3 | 92.3±0.2 | 93.1±0.1 | 95.4±0.3 | 97.0±0.2 |
| VBMTL | 90.8±0.6 | 93.2±0.2 | 93.5±0.1 | 96.5±0.1 | 97.1±0.4 |
| VMTL-AC | **93.9±0.1** | 95.1±0.0 | **95.2±0.1** | **96.8±0.2** | 97.2±0.2 |
| VMTL | 93.8±0.1 | **95.3±0.0** | **95.2±0.1** | 96.5±0.2 | **97.3±0.1** |

Table C.6. Performance under different proportions of training data on *ImageCLEF*.

| Methods | 5% | 10% | 20% | 40% | 60% |
|---|---|---|---|---|---|
| STL | 62.6±0.2 | 69.7±0.3 | 76.2±0.3 | 79.3±0.2 | 80.1±0.1 |
| VSTL | 64.9±0.3 | 70.8±0.3 | 77.2±0.2 | 80.3±0.1 | 80.8±0.1 |
| BMTL | 65.7±0.4 | 72.0±0.3 | 76.8±0.3 | 79.0±0.3 | 80.5±0.2 |
| VBMTL | 67.1±0.3 | 73.0±0.7 | 78.0±0.2 | 81.2±0.1 | 81.0±0.2 |
| VMTL-AC | 75.7±0.3 | 77.6±0.2 | 80.0±0.1 | 81.3±0.2 | **82.3±0.3** |
| VMTL | **76.2±0.3** | **77.9±0.2** | **80.2±0.1** | **82.4±0.4** | 82.3±0.2 |

variational Bayesian framework. Given a limited amount of training data, our models also have a better performance than VBMTL, which demonstrates that the proposed Gumbel-Softmax priors are beneficial to fully leverage the shared knowledge among tasks. When training data is limited, STL and VSTL can not train a proper model for each task. As the training data decreases, our methods based on the variational Bayesian framework are able to better handle this challenging case by incorporating the shared knowledge into the prior of each task. The best and second-best results of average accuracy are respectively marked in bold and underlined.

## C.2 Effectiveness of Variational Bayesian Approximation

Comparative results on performance of Bayesian approximation for representations **z** and classifiers **w** on the *Office-Home*, *Office-Caltech* and *ImageCLEF* datasets are shown in Tables C.7, C.8 and C.9, respectively. Both variational Bayesian representations and classifiers can benefit performance. Our method jointly infers the posteriors over feature representations and classifiers in a Bayesian framework and outperforms its variants on three benchmarks for most of train-test splits, which demonstrates the benefits of applying Bayesian inference to both classifiers and representations.

## C.3 Effectiveness of Gumbel-Softmax Priors

The performance comparison of the proposed VMTL with different priors on the *Office-Home*, *Office-Caltech* and *ImageCLEF* datasets are shown in Tables C.10, C.11 and C.12, respectively. These approximated priors are obtained by combining posteriors of related tasks. "Mean" denotes that the prior of the current task is the mean of variational posteriors of other related tasks. "Learnable weighted" denotes that weights of mixing the variational posteriors of other related tasks are learnable. Our proposed prior by the Gumbel-Softmax technique to learn the mixing weights introduces uncertainty to the relationships among tasks to explore sufficient transferable information from other

Table C.7. Detailed results on performance of Bayesian approximation for representation **z** and classifier **w** on *Office-Home*.

| z | w | 5% | | | | | 10% | | | | | 20% | | | | |
|---|---|---|---|---|---|---|---|---|---|---|---|---|---|---|---|---|
| | | A | C | P | R | Avg. | A | C | P | R | Avg. | A | C | P | R | Avg. |
| ✗ | ✗ | $37.6_{\pm0.4}$ | $31.5_{\pm0.3}$ | $68.5_{\pm0.2}$ | $63.8_{\pm0.2}$ | $50.4_{\pm0.1}$ | $51.0_{\pm0.2}$ | $41.6_{\pm0.1}$ | $76.0_{\pm0.4}$ | $69.2_{\pm0.3}$ | $59.5_{\pm0.1}$ | $56.6_{\pm0.3}$ | $51.8_{\pm0.5}$ | $80.9_{\pm0.4}$ | $72.9_{\pm0.4}$ | $65.6_{\pm0.1}$ |
| ✓ | ✗ | $52.0_{\pm0.4}$ | $37.6_{\pm0.3}$ | $71.8_{\pm0.4}$ | $68.3_{\pm0.1}$ | $57.4_{\pm0.0}$ | $59.2_{\pm0.4}$ | $47.1_{\pm0.2}$ | $77.6_{\pm0.1}$ | $73.7_{\pm0.4}$ | $64.4_{\pm0.1}$ | $63.3_{\pm0.4}$ | $52.7_{\pm0.2}$ | $82.1_{\pm0.1}$ | $76.0_{\pm0.3}$ | $68.5_{\pm0.1}$ |
| ✗ | ✓ | $51.6_{\pm0.3}$ | $37.8_{\pm0.1}$ | $70.7_{\pm0.4}$ | $66.9_{\pm0.2}$ | $56.8_{\pm0.1}$ | $57.6_{\pm0.3}$ | $47.2_{\pm0.2}$ | $77.4_{\pm0.3}$ | $72.5_{\pm0.1}$ | $63.7_{\pm0.1}$ | $62.5_{\pm0.4}$ | $53.5_{\pm0.3}$ | $82.0_{\pm0.2}$ | $76.0_{\pm0.3}$ | $68.5_{\pm0.1}$ |
| ✓ | ✓ | $53.5_{\pm0.1}$ | $39.2_{\pm0.2}$ | $71.5_{\pm0.3}$ | $69.0_{\pm0.1}$ | $\mathbf{58.3_{\pm0.1}}$ | $60.2_{\pm0.3}$ | $47.5_{\pm0.2}$ | $78.2_{\pm0.3}$ | $74.0_{\pm0.2}$ | $\mathbf{65.0_{\pm0.0}}$ | $64.0_{\pm0.3}$ | $53.5_{\pm0.4}$ | $82.5_{\pm0.2}$ | $76.8_{\pm0.1}$ | $\mathbf{69.2_{\pm0.2}}$ |

Table C.8. Detailed results on performance of Bayesian approximation for representation **z** and classifier **w** on *Office-Caltech*.

| z | w | 5% | | | | | 10% | | | | | 20% | | | | |
|---|---|---|---|---|---|---|---|---|---|---|---|---|---|---|---|---|
| | | A | W | D | C | Avg. | A | W | D | C | Avg. | A | W | D | C | Avg. |
| ✗ | ✗ | $90.0_{\pm0.7}$ | $89.4_{\pm0.8}$ | $95.0_{\pm1.1}$ | $83.5_{\pm0.5}$ | $89.5_{\pm0.3}$ | $93.6_{\pm0.1}$ | $97.0_{\pm0.6}$ | $92.1_{\pm0.7}$ | $86.3_{\pm0.4}$ | $92.3_{\pm0.2}$ | $95.0_{\pm0.2}$ | $94.5_{\pm0.7}$ | $96.0_{\pm1.2}$ | $86.8_{\pm0.2}$ | $93.1_{\pm0.1}$ |
| ✓ | ✗ | $93.3_{\pm0.4}$ | $95.0_{\pm0.5}$ | $96.1_{\pm0.3}$ | $90.0_{\pm0.3}$ | $93.6_{\pm0.2}$ | $95.3_{\pm0.1}$ | $97.4_{\pm0.3}$ | $97.9_{\pm0.0}$ | $90.4_{\pm0.3}$ | $95.2_{\pm0.0}$ | $95.6_{\pm0.1}$ | $96.6_{\pm0.5}$ | $98.4_{\pm0.4}$ | $90.3_{\pm0.6}$ | $95.2_{\pm0.2}$ |
| ✗ | ✓ | $93.2_{\pm0.1}$ | $95.5_{\pm0.3}$ | $95.7_{\pm0.5}$ | $89.6_{\pm0.2}$ | $93.5_{\pm0.1}$ | $94.8_{\pm0.2}$ | $97.3_{\pm0.4}$ | $97.8_{\pm0.4}$ | $90.6_{\pm0.3}$ | $95.1_{\pm0.1}$ | $95.6_{\pm0.1}$ | $96.6_{\pm0.3}$ | $99.2_{\pm0.3}$ | $89.8_{\pm0.4}$ | $\mathbf{95.3_{\pm0.2}}$ |
| ✓ | ✓ | $93.7_{\pm0.2}$ | $95.2_{\pm0.1}$ | $96.4_{\pm0.4}$ | $89.7_{\pm0.4}$ | $\mathbf{93.8_{\pm0.1}}$ | $95.4_{\pm0.1}$ | $97.6_{\pm0.1}$ | $97.4_{\pm0.4}$ | $90.9_{\pm0.2}$ | $\mathbf{95.3_{\pm0.0}}$ | $95.6_{\pm0.2}$ | $95.6_{\pm0.4}$ | $98.4_{\pm0.5}$ | $91.1_{\pm0.3}$ | $95.2_{\pm0.1}$ |

Table C.9. Detailed results on performance of Bayesian approximation for representation **z** and classifier **w** on *ImageCLEF*.

| z | w | 5% | | | | | 10% | | | | | 20% | | | | |
|---|---|---|---|---|---|---|---|---|---|---|---|---|---|---|---|---|
| | | C | I | P | B | Avg. | C | I | P | B | Avg. | C | I | P | B | Avg. |
| ✗ | ✗ | $88.3_{\pm0.5}$ | $73.2_{\pm0.5}$ | $61.2_{\pm0.9}$ | $40.0_{\pm0.8}$ | $65.7_{\pm0.4}$ | $90.6_{\pm0.8}$ | $79.3_{\pm0.2}$ | $66.3_{\pm0.5}$ | $51.9_{\pm0.6}$ | $72.0_{\pm0.3}$ | $92.9_{\pm0.5}$ | $86.5_{\pm0.2}$ | $71.9_{\pm0.6}$ | $56.0_{\pm0.8}$ | $76.8_{\pm0.3}$ |
| ✓ | ✗ | $90.4_{\pm0.2}$ | $81.9_{\pm0.7}$ | $70.9_{\pm0.5}$ | $57.9_{\pm0.6}$ | $75.3_{\pm0.4}$ | $93.7_{\pm0.2}$ | $85.7_{\pm0.6}$ | $71.7_{\pm0.2}$ | $59.1_{\pm0.3}$ | $77.5_{\pm0.0}$ | $93.5_{\pm0.2}$ | $89.8_{\pm0.6}$ | $77.3_{\pm0.5}$ | $59.0_{\pm0.4}$ | $79.9_{\pm0.1}$ |
| ✗ | ✓ | $91.4_{\pm1.1}$ | $81.9_{\pm0.5}$ | $71.8_{\pm0.6}$ | $58.4_{\pm0.5}$ | $75.9_{\pm0.2}$ | $93.0_{\pm0.6}$ | $86.3_{\pm0.4}$ | $72.0_{\pm0.6}$ | $60.0_{\pm0.6}$ | $77.8_{\pm0.3}$ | $93.8_{\pm0.5}$ | $88.1_{\pm0.5}$ | $77.1_{\pm0.8}$ | $59.0_{\pm0.4}$ | $79.5_{\pm0.2}$ |
| ✓ | ✓ | $91.5_{\pm0.2}$ | $83.5_{\pm0.6}$ | $71.6_{\pm0.8}$ | $58.2_{\pm0.7}$ | $\mathbf{76.2_{\pm0.3}}$ | $94.2_{\pm0.2}$ | $86.0_{\pm0.5}$ | $71.7_{\pm0.6}$ | $59.8_{\pm0.4}$ | $\mathbf{77.9_{\pm0.2}}$ | $93.9_{\pm0.1}$ | $89.4_{\pm0.2}$ | $78.0_{\pm0.4}$ | $59.5_{\pm0.4}$ | $\mathbf{80.2_{\pm0.1}}$ |

Table C.10. Detailed results on performance of VMTL with different priors on *Office-Home*.

| Priors | 5% | | | | | 10% | | | | | 20% | | | | |
|---|---|---|---|---|---|---|---|---|---|---|---|---|---|---|---|
| | A | C | P | R | Avg. | A | C | P | R | Avg. | A | C | P | R | Avg. |
| Mean | $52.2_{\pm0.4}$ | $38.0_{\pm0.2}$ | $71.3_{\pm0.3}$ | $68.3_{\pm0.1}$ | $57.5_{\pm0.0}$ | $59.1_{\pm0.4}$ | $47.1_{\pm0.2}$ | $77.6_{\pm0.1}$ | $73.7_{\pm0.4}$ | $64.4_{\pm0.1}$ | $63.3_{\pm0.4}$ | $52.7_{\pm0.2}$ | $82.1_{\pm0.1}$ | $76.0_{\pm0.3}$ | $68.5_{\pm0.1}$ |
| Learnable weighted | $51.8_{\pm0.5}$ | $38.0_{\pm0.4}$ | $70.9_{\pm0.5}$ | $67.9_{\pm0.4}$ | $57.2_{\pm0.2}$ | $59.2_{\pm0.5}$ | $46.9_{\pm0.4}$ | $77.6_{\pm0.3}$ | $73.2_{\pm0.3}$ | $64.2_{\pm0.2}$ | $63.9_{\pm0.5}$ | $53.2_{\pm0.4}$ | $82.1_{\pm0.4}$ | $76.3_{\pm0.2}$ | $68.9_{\pm0.2}$ |
| Gumbel-Softmax | $53.5_{\pm0.1}$ | $39.2_{\pm0.2}$ | $71.5_{\pm0.3}$ | $69.0_{\pm0.1}$ | $\mathbf{58.3_{\pm0.1}}$ | $60.2_{\pm0.2}$ | $47.5_{\pm0.2}$ | $78.2_{\pm0.3}$ | $74.0_{\pm0.2}$ | $\mathbf{65.0_{\pm0.0}}$ | $64.0_{\pm0.3}$ | $53.5_{\pm0.4}$ | $82.5_{\pm0.2}$ | $76.8_{\pm0.1}$ | $\mathbf{69.2_{\pm0.2}}$ |

Table C.11. Detailed results on performance of VMTL with different priors on *Office-Caltech*.

| Priors | 5% | | | | | 10% | | | | | 20% | | | | |
|---|---|---|---|---|---|---|---|---|---|---|---|---|---|---|---|
| | A | W | D | C | Avg. | A | W | D | C | Avg. | A | W | D | C | Avg. |
| Mean | $93.3_{\pm0.4}$ | $95.0_{\pm0.5}$ | $96.1_{\pm0.3}$ | $90.0_{\pm0.3}$ | $93.6_{\pm0.2}$ | $95.1_{\pm0.4}$ | $97.4_{\pm0.3}$ | $97.9_{\pm0.0}$ | $90.7_{\pm0.6}$ | $\mathbf{95.3_{\pm0.0}}$ | $95.6_{\pm0.2}$ | $96.1_{\pm0.5}$ | $98.7_{\pm0.4}$ | $91.0_{\pm0.6}$ | $\mathbf{95.4_{\pm0.2}}$ |
| Learnable weighted | $93.6_{\pm0.3}$ | $95.0_{\pm0.2}$ | $96.7_{\pm0.7}$ | $89.4_{\pm0.7}$ | $93.7_{\pm0.2}$ | $94.9_{\pm0.3}$ | $97.3_{\pm0.6}$ | $97.9_{\pm0.0}$ | $90.4_{\pm0.3}$ | $95.1_{\pm0.1}$ | $95.6_{\pm0.2}$ | $95.9_{\pm0.3}$ | $98.2_{\pm0.8}$ | $90.9_{\pm0.6}$ | $95.1_{\pm0.1}$ |
| Gumbel-Softmax | $93.7_{\pm0.2}$ | $95.2_{\pm0.1}$ | $96.4_{\pm0.4}$ | $89.7_{\pm0.4}$ | $\mathbf{93.8_{\pm0.1}}$ | $95.4_{\pm0.1}$ | $97.6_{\pm0.1}$ | $97.4_{\pm0.4}$ | $90.9_{\pm0.2}$ | $\mathbf{95.3_{\pm0.0}}$ | $95.6_{\pm0.2}$ | $95.6_{\pm0.4}$ | $98.4_{\pm0.5}$ | $91.1_{\pm0.3}$ | $95.2_{\pm0.1}$ |

Table C.12. Detailed results on performance of VMTL with different priors on *ImageCLEF*.

| Priors | 5% | | | | | 10% | | | | | 20% | | | | |
|---|---|---|---|---|---|---|---|---|---|---|---|---|---|---|---|
| | C | I | P | B | Avg. | C | I | P | B | Avg. | C | I | P | B | Avg. |
| Mean | $90.4_{\pm0.2}$ | $82.7_{\pm0.7}$ | $71._{\pm0.5}$ | $57.8_{\pm0.6}$ | $75.5_{\pm0.4}$ | $93.4_{\pm0.2}$ | $86.4_{\pm0.4}$ | $71.5_{\pm0.2}$ | $59.1_{\pm0.3}$ | $77.6_{\pm0.0}$ | $93.6_{\pm0.2}$ | $89.7_{\pm0.6}$ | $77.8_{\pm0.5}$ | $59.4_{\pm0.4}$ | $80.1_{\pm0.1}$ |
| Learnable weighted | $90.1_{\pm0.4}$ | $82.1_{\pm0.2}$ | $71.2_{\pm1.0}$ | $58.5_{\pm0.6}$ | $75.5_{\pm0.4}$ | $93.4_{\pm0.3}$ | $85.6_{\pm0.5}$ | $71.5_{\pm0.5}$ | $59.1_{\pm0.6}$ | $77.4_{\pm0.2}$ | $93.5_{\pm0.3}$ | $89.5_{\pm0.2}$ | $77.9_{\pm0.7}$ | $59.6_{\pm0.4}$ | $80.1_{\pm0.2}$ |
| Gumbel-Softmax | $91.5_{\pm0.2}$ | $83.5_{\pm0.6}$ | $71.6_{\pm0.8}$ | $58.2_{\pm0.7}$ | $\mathbf{76.2_{\pm0.3}}$ | $94.2_{\pm0.2}$ | $86.0_{\pm0.5}$ | $71.7_{\pm0.6}$ | $59.8_{\pm0.4}$ | $\mathbf{77.9_{\pm0.2}}$ | $93.9_{\pm0.1}$ | $89.4_{\pm0.2}$ | $78.0_{\pm0.4}$ | $59.5_{\pm0.4}$ | $\mathbf{80.2_{\pm0.1}}$ |

Table C.13. Performance comparison of different methods on the large-scaled dataset *DomainNet* for multiple tasks: Clipart (C), Infograph (I), Painting (P), Quickdraw (Q), Real (R) and Sketch (S).

| Methods | 4% | | | | | | |
|---|---|---|---|---|---|---|---|
| | C | I | P | Q | R | S | Avg. |
| STL | $23.0_{\pm0.2}$ | $7.1_{\pm0.3}$ | $30.4_{\pm0.1}$ | $5.2_{\pm0.2}$ | $58.7_{\pm0.3}$ | $16.3_{\pm0.3}$ | $23.5_{\pm0.2}$ |
| VSTL | $33.8_{\pm0.1}$ | $12.3_{\pm0.2}$ | $37.1_{\pm0.1}$ | $23.7_{\pm0.3}$ | $65.3_{\pm0.4}$ | $23.2_{\pm0.2}$ | $32.6_{\pm0.1}$ |
| Bakker et al. [3] | $28.3_{\pm0.4}$ | $10.7_{\pm0.2}$ | $35.4_{\pm0.1}$ | $22.4_{\pm0.3}$ | $59.9_{\pm0.3}$ | $21.7_{\pm0.3}$ | $29.7_{\pm0.3}$ |
| Long et al. [33] | $28.7_{\pm0.1}$ | $13.2_{\pm0.3}$ | $38.7_{\pm0.4}$ | $7.5_{\pm0.3}$ | $62.9_{\pm0.2}$ | $20.6_{\pm0.3}$ | $28.6_{\pm0.2}$ |
| Kendall et al. [25] | $30.2_{\pm0.3}$ | $11.6_{\pm0.4}$ | $37.3_{\pm0.4}$ | $29.7_{\pm0.2}$ | $62.1_{\pm0.5}$ | $22.2_{\pm0.3}$ | $32.2_{\pm0.3}$ |
| Qian et al. [38] | $32.5_{\pm0.3}$ | $12.6_{\pm0.2}$ | $40.4_{\pm0.4}$ | $25.8_{\pm0.5}$ | $64.4_{\pm0.2}$ | $25.4_{\pm0.3}$ | $33.5_{\pm0.3}$ |
| BMTL | $34.0_{\pm0.1}$ | $11.9_{\pm0.3}$ | $36.8_{\pm0.1}$ | $24.7_{\pm0.2}$ | $64.9_{\pm0.2}$ | $23.1_{\pm0.3}$ | $32.6_{\pm0.1}$ |
| VBMTL | $33.1_{\pm0.1}$ | $12.0_{\pm0.2}$ | $37.0_{\pm0.2}$ | $19.7_{\pm0.1}$ | $64.5_{\pm0.1}$ | $22.8_{\pm0.3}$ | $31.5_{\pm0.1}$ |
| **VMTL-AC** | $31.4_{\pm0.1}$ | $11.1_{\pm0.1}$ | $35.3_{\pm0.1}$ | $15.8_{\pm0.1}$ | $61.5_{\pm0.1}$ | $21.8_{\pm0.2}$ | $29.5_{\pm0.1}$ |
| **VMTL** | $36.4_{\pm0.2}$ | $14.8_{\pm0.2}$ | $40.0_{\pm0.1}$ | $19.0_{\pm0.1}$ | $65.5_{\pm0.1}$ | $26.1_{\pm0.1}$ | $\mathbf{33.6_{\pm0.1}}$ |

Table C.14. Impact of $L$ and $M$ on performance. Experiments are conducted on *Office-Home* with a 5% train-test split.

| L (=M) | 1 | 10 | 20 | 30 | 40 | 50 | 60 | 70 | 80 | 90 | 100 |
|---|---|---|---|---|---|---|---|---|---|---|---|
| **VMTL-AC** | 56.1 | **56.3** | 56.1 | 56.2 | 56.3 | 56.1 | **56.3** | 56.0 | 56.2 | **56.3** | 55.8 |
| **VMTL** | 58.1 | **58.3** | 58.2 | 58.2 | 58.2 | **58.3** | 58.0 | **58.3** | 58.2 | 58.2 | 58.2 |

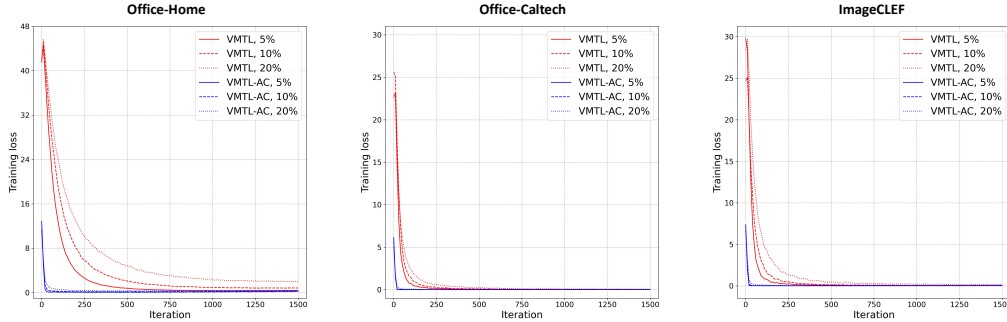

Fig. C.1. Illustration of training loss with iterations on *Office-Home*, *Office-Caltech* and *ImageCLEF*. VMTL-AC converges faster than VMTL under 5%, 10% and 20% train-test splits, which demonstrates the computational benefit of amortized learning.

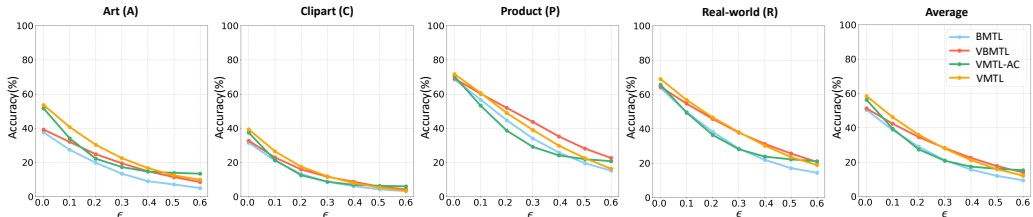

Fig. C.2. The performance for each task under different noise levels on the *Office-Home* dataset.

tasks. In the three datasets, our designed priors outperform other methods under most of the train-test splits. In addition, the results of *DomainNet* under the 4% train-test split are provided in Table C.13.

We learn different weights for $\mathbf{w}$ and $\mathbf{z}$, motivated by the fact that latent variables are different in terms of capturing uncertainty: $\mathbf{w}$ is at the category level while $\mathbf{z}$ at the instance level. To validate, we conduct experiments using the same Gumbel-Softmax weights for both variables. As shown in Table C.15, using different weights are slightly better than using the same Gumbel-Softmax weights.

Table C.15. Performance of our priors using the same weights for $\mathbf{w}$ and $\mathbf{z}$ or not on *Office-Home*.

| Train-test split | 5% | 10% | 20% |
|---|---|---|---|
| Same weights | 58.2±0.1 | 64.8±0.1 | 68.7±0.3 |
| Different weights | **58.3±0.1** | **65.0±0.0** | **69.2±0.2** |

## C.4 Sensitivity of the Hyper-parameter $L$ and $M$

In this paper, $L$ and $M$ are the number of Monte Carlo samples for the variational posteriors of latent representations and classifiers, respectively. We ablate the sensitivity of $L$ and $M$ in Table C.14 on the *Office-Home* dataset. In practice, $L$ and $M$ are set to ten, which offer a good balance between accuracy and efficiency.

## C.5 Fast Convergence of Amortized Classifiers

The computational advantage of amortized classifiers can be illustrated by the training loss as function of iteration on *Office-Home*, *Office-Caltech* and *ImageCLEF*. As shown in Fig. C.1, VMTL-AC converges faster than VMTL under 5%, 10% and 20% train-test splits, which demonstrates the computational benefit of amortized learning.

## C.6 Robustness of Our Methods

We conduct experiments on the *Office-home* dataset to show the robustness of our methods against adversarial attacks. In our experiments, the adversarial attack is implemented by the fast gradient sign method [**?** ] where $\epsilon$ denotes the noise level. As shown in Fig. C.2, under different noise levels, the proposed model VMTL outperforms BMTL. As the noise level increases, the proposed model VMTL-AC is more robust than other models.

## C.7 A New Metric for Evaluating the Uncertainty Prediction

For Bayesian methods, it is necessary to quantify the model's ability of handling uncertainty. We looked into related references and didn't find such a measure for comparing the uncertainty prediction. Thus, we adopt a new metric for evaluating the uncertainty prediction, the ratio of the average entropy of failure cases and properly classified samples. If the ratio is higher, the Bayesian methods predict failure cases with more uncertainty and predict successful cases with more confidence. As shown in Table C.16, VMTL has higher entropy ratios, which demonstrates the effectiveness of our model to handle the uncertainty.

Table C.16.   Entropy ratio (the higher the better) on *Office-Home*.

| Train-test split | 5% | 10% | 20% |
|---|---|---|---|
| Bakker et al.[3] | 2.469 | 2.625 | 3.031 |
| VBMTL | 4.111 | 4.430 | 5.460 |
| **VMTL** | **4.546** | **4.472** | **5.584** |

## C.8 Runtime Impact of the Sampling Steps

To investigate the runtime impact of the additional sampling steps we compare the actual training and inference time of the proposed method with that of deterministic approaches. As shown in Table C.17, compared to the deterministic baseline (BMTL), the training and inference time of our method increases as the number of MC samples is set higher. In this paper, the number of MC samples is set to be 10, which is computationally efficient while yielding good performance (Table C.14). In this case, our method does cost extra at training time but with 0.122s per iteration, this is still acceptable. When testing 1000 samples, our method only increases by an extra 10% test time of BMTL. Thus, our model doesn't cost much more time to surpass BMTL by 7.9% in terms of accuracy.

Table C.17.  Runtime impact (seconds) of sampling step on Office-Home with 5% split.

| Methods | BMTL | VMTL | | | |
|---|---|---|---|---|---|
| MC samples | - | 1 | 10 | 50 | 100 |
| Training (per iteration) | 0.040 | 0.098 | 0.122 | 0.197 | 0.320 |
| Inference (per 1000 test samples) | 0.325 | 0.343 | 0.357 | 0.371 | 0.426 |