# OpenReview forum: "Variational Multi-Task Learning with Gumbel-Softmax Priors"
_NeurIPS.cc/2021/Conference — NeurIPS 2021 Poster_

### Official Review · Reviewer_xbXZ · 2021-07-10

**Rating:** 5
**Confidence:** 3

**Summary:**

This work proposes a probabilistic method (variational inference) to exploit task relatedness and ignore harmful interference in multi-task learning paradigms. In particular, the authors focus on a unique MTL setting (training data is extremely limited; tasks have different input spaces; tasks have the same target space) to highlight the utility of their method. Experimental results on 5 datasets and compared to 6 related work is highly compelling to establishing the efficacy of the proposed method.

**Limitations And Societal Impact:**

I cannot recall this work containing meaningful discussion of its limitations (aside from the limited scope of its applicability) and societal impact is not addressed.

**Main Review:**

This work is grounded in the probabilistic multi-task learning domain but presents a novel approach: using variational inference to model task relatedness. This method appears highly effective when training data is extremely limited as it augments each task with a strong inductive bias by learning correlation patterns among tasks (Figure 3). Moreover, these correlation patterns appear to be sparse (good) which the authors attribute to using a Gumbel-Softmax to determine task relatedness.

The submission is clearly written and cogent. The authors motivate their proposed method, outline the situation in which it is useful, offer intuition and explanation in defining methodology and comprehensively present compelling experimental results which adhere to their introduction. I believe the approach is technically sound; although, I did not carefully check for conflicting assumptions or unsupported jumps in the methodology. The extensive empirical evaluation significantly bolsters the author's proposed method.

That said, the work addresses a very limited use case of MTL, and I have not encountered a use case -- and have difficulty imagining a real world scenario -- where this technique would be used. The datasets used in the empirical evaluation are quite unusual: "small" datasets with an even smaller 0.1% -> 20% of the available training data being used for the method to be effective. Moreover, I would posit a transfer learning algorithm or meta learning algorithm would be significantly more suitable to this paradigm than a multi-task learning approach. Additionally, the method is quite involved and complex which reduces its utility and adoption. With regards to presenting novel research concepts, this work suggests their contribution is jointly exploring learned representations and classifiers; however, I would suggest the domain of soft-parameter sharing and even neural architecture search in MTL systems presently address this domain, but with significantly more utility on a variety of MTL applications. Integrating variational inference into multi-task learning is novel, but again, the extensibility of this contribution and application to MTL systems seems extremely limited.

In summary, this is a very high quality and well-written work, but I struggle to see the significance of the proposed method to either real world systems or as a building block in future work.

**Time Spent Reviewing:**

2

---

> ### Author Response · Authors · 2021-08-10
> **Response to Reviewer xbXZ**
>
> We thank Reviewer xbXZ for the positive comments "presents a novel approach", "the submission is clearly written and cogent" and "this is a very high quality and well-written work".
>
> **Q:** *In summary, this is a very high quality and well-written work, but I struggle to see the significance of the proposed method to either real world systems or as a building block in future work.*
>
> **A:** We thank the reviewer for mentioning this. We would be happy to take this opportunity to discuss the significance of the proposed method in the following several aspects.
>
> (1) Our method has a high potential to improve knowledge sharing between different data distributions from different modalities. For example, in self-driving cars, the fine-grained categorization of the driver behaviors needs multi-modal activity recognition to be done in real-time with input data from multi-view and multi-modality [a].
>
> We note that our method not only shows dominant performance in the setting with limited data, but also produces comparable even better performance than baselines in the setting with large amounts of data, as shown in Fig.2 and the second table of the response to Reviewer Q1H6. In this sense, our method is not restricted to the setting that training data is extremely limited.
>
> Moreover, as a matter of fact, our work is a typical transfer learning. Our method promotes positive transferring and reduces negative transferring among tasks by applying the proposed Gumbel-Softmax prior. We agree meta-learning is effective to handle limited data. However, it would not be feasible to posit meta learning here because in this setting we only have several tasks to gain shared knowledge, while meta learning usually relies on hundreds of training tasks that are organized from large amounts of meta-data [b].
>
> (2) In our method, designing the prior by aggregating variational posteriors from related tasks is not limited to a specific setting. It can work as a building block in future work, which is typical for soft-parameter sharing in MTL systems.
> This design provides a general mechanism of knowledge sharing to handle the cross-task talk [c] in a common latent space. This design already works as a building block in our method, which is applied to a feature extraction layer and the classifier layer to utilize the shared knowledge among related tasks.
>
> [a] Martin, Manuel, et al. Drive\&act: A multi-modal dataset for fine-grained driver behavior recognition in autonomous vehicles, Proceedings of the IEEE CVF International Conference on Computer Vision, 2019. (https://openaccess.thecvf.com/content_ICCV_2019/papers/Martin_DriveAct_A_Multi-Modal_Dataset_for_Fine-Grained_Driver_Behavior_Recognition_in_ICCV_2019_paper.pdf)
>
> [b] Vanschoren, Joaquin. Meta-learning: A survey, arXiv preprint arXiv:1810.03548, 2018. (https://arxiv.org/pdf/1810.03548.pdf)
>
> [c] Vandenhende, Simon, et al. Multi-task learning for dense prediction tasks: A survey, IEEE Transactions on Pattern Analysis and Machine Intelligence, 2021. (https://arxiv.org/pdf/2004.13379.pdf)
>
> **Limitations and societal impact:**
>
> (1) As indicated for Reviewer Q1H6 when there is ample training data our method in isolation doesn't yield a prominent performance increase. It would need to be combined with deterministic approaches to extract more representative features specifically for individual tasks.  We will add those discussions about the limitation in the paper.
>
> (2) Our method is a general contribution to the MTL field and has the potential to benefit any societal problem where multiple tasks are performed simultaneously ranging from medical diagnosis to autonomous driving. Thank you.

---

### Official Review · Reviewer_TXRx · 2021-07-13

**Rating:** 6
**Confidence:** 4

**Summary:**

This paper proposes a unified few-shot multi-task learning framework which learns latent representations of data points and latent classifier weights through an amortized variational inference. Specifically, the proposed method regularizes the inference networks by minimizing KL divergence between the target variational distribution and the mixture of the non-target variational distribution. The mixing probability of the non-target variational distribution is sampled through Gumbel-Softmax distribution with learnable parameters. The intuition behind this regularization is that a model is able to learn from related tasks to improve the inference of latent representations and classifier weights (especially useful for the few-shot multi-task learning setting). The extensive experiments demonstrates that the proposed method is more effective than other methods for the few-shot multi-task learning.

**Limitations And Societal Impact:**

The below is my and questions and suggestions to improve this paper.

- Although the idea of using an aggregated (or mixture of) variational posterior as a prior distribution is new for multi-task learning, it is already introduced in the literature of improving Variational Autoencoder (https://arxiv.org/pdf/1809.05284.pdf). It would be better if any discussion of this paper is included in the paper.

- This paper regularizes a model to use the data of not-target tasks by minimizing KL divergence between the variational distribution of target tasks and the mixture of the non-target distributions. I am not enthusiastic for the theory, but as far as I know any analytic solution of KL divergence between Gaussian distribution and a mixture of Gaussian distributions does not exists, and only bound exists. Can you elaborate how to compute KL divergence in your framework?

- The function f is introduced in equation 10, however, its usage is not explained in relation to equation 9. Is this function q_\phi in equation 9?

- I didn't understand how to select the class c in equation 10. Is this class is the same class of x_{t, n}? if it is, how it work when we do not know the label of x_{t, n} as equation 9 and 10?

- I have checked the implementation details of supplementary files to understand the above question. As I read, the cross attention f in equation 10 is used for the prior distribution of the latent representation, however, the cross attention layer appears only for the inference network \theta of classifier weights (in Table B.3). This is very different as I read in the paper.


**Main Review:**

- Originality

This paper tackles the uncertainty of few-shot multi-task learning through a unified probabilistic inference of representation and classifier weights, which is novel in the literature of few-shot multi-task learning. The difference from the related work is also well-explained in Section 3.

- Quality

The technical quality of this paper is intuitive and well-demonstrated, however, there is some missing part for technical details, such as, KL regularization (see the below questions).  The experimental results are very impressive and coherent to the motivation and design purpose, for example, the proposed method is most useful for the most limited data setting.

- Clarity

This paper is well-written, but there are equations that can lead misunderstandings.

---
I appreciate the authors to address my concern, and I will raise my score.




**Time Spent Reviewing:**

5 hours

---

> ### Author Response · Authors · 2021-08-10
> **Response to Reviewer TXRx**
>
> We thank Reviewer TXRx for stating our proposal is novel, intuitive and well-demonstrated; the experimental results are very impressive and coherent; and this paper is well-written.
>
> 1. We thank you for bringing this reference [a], which we will include in our paper for discussion. The idea of designing informative priors by aggregating posterior is in a similar spirit to our work while the technical implementations and problem settings are fundamentally different. We summarize the major differences below for your quick reference which will be added to the paper.
>
>     *  [a] estimates the KL term with an introduced probabilistic binary classifier, while our method adopts a closed form based on minimizing the upper bound of the KL term.
>
>     *  [a] treats posteriors equally in aggregation, while our method adopts fully learnable weights by applying the Gumbel-Softmax technique.
>
>     *  [a] is developed in the VAE framework to improve it for generative modeling, our method is designed for exploring task relationships in supervised multi-task learning.
>
>       [a] Takahashi, Hiroshi, et al. Variational autoencoder with implicit optimal priors, Proceedings of the AAAI Conference on Artificial Intelligence, 2019. (https://arxiv.org/pdf/1809.05284.pdf)
>
> 2. Thank you for this insightful comment. In the practical implementation of the KL term, we adopt the closed-form solution based on its upper bound as done in [b]: $D_{KL} \big[ q_{\theta} (w_t) || \sum_{t \neq i} \alpha_{ti}  q_\theta(w_i) \big] \leq \sum_{t \neq i} \alpha_{ti} D_{KL} \big[ q_\theta(w_t) ||  q_\theta(w_i) \big]$. We apologize for not explicitly mentioning this and will elaborate it in the paper.
>
>     [b] Nalisnick E, Smyth P. Learning priors for invariance, International Conference on Artificial Intelligence and Statistics, 2018. (http://proceedings.mlr.press/v84/nalisnick18a/nalisnick18a.pdf)
>
> 3. Thank you for this comment and we are sorry for the caused confusion here. $f$ is a function that aggregates the conditionals of $q_\phi$ (as mentioned in ln. 153). The inference network takes the aggregated representation as input and returns the parameters of the distribution $q_\phi$. We will revise this part to make it clearer.
>
> 4. You are correct. In Eq.(10), c is the same class of $x_{t, n}$. Since we are dealing with supervised learning in this work, class labels of training samples are always available at training time. During test time, the varitional posterior is used to infer the distributions of the latent variables without using the label of the test data. Your mentioned scenario when we do not know the label could happen in other settings, e.g., semi-supervised learning, where we may have unlabelled samples. In this case, we can find similar samples of $x_{t, n}$ in $\mathcal{D}_i$ by measuring their distance. In any case, the rationale is to find similar samples to help build the representation of the current sample. We will also discuss this in the paper. Thank you.
>
> 5. We apologize for the caused confusion here. The two captions of Tables B.2 and B.3 were mistaken. Actually, Table B.3 describes the architecture of the inference network $\phi$ for latent representations, and Table B.2 is for classifier weights $\theta$. Therefore, the cross attention layer appears only for representations. We will correct this. Thank you.

---

### Official Review · Reviewer_Q1H6 · 2021-07-13

**Rating:** 7
**Confidence:** 4

**Summary:**

In this work, the authors propose to use Gumbel-Softmax as the prior of a task for multi-task learning under the variational Bayesian inference setup. Instead of exploiting task relatedness for representations only, the proposed method jointly infers posteriors for both representations and classifiers. State-of-the-art results are achieved in the multi-input multi-output setting with a small training data regime.

**Limitations And Societal Impact:**

The main limitation I can think of is the performance of the proposed variational Bayesian approach in the large data regime as mentioned above. I was wondering if there is a way to combine the advantages of the proposed approach with deterministic approaches so that it can generalize well in more settings?

**Main Review:**

Strengths:
* Thorough literature review in terms of deterministic and probabilistic approaches.
* Leverage the advantages of probabilistic-based models and achieve good results on the small data regime.
* Various ablation studies demonstrate the effectiveness of Gumbel-Softmax in MTL environments and effectiveness of the proposed variational Bayesian representation z and classifier w.
* Although Gumbel-Softmax has been widely used in multi-task learning [7,19,42], to the best of my knowledge it is the first time being applied to the variational multi-task learning setup.

Weaknesses:
* It would be great to discuss the actual training and inference time of the proposed method compared to deterministic approaches to see the runtime impact of the additional sampling steps.
* The benefit of a probabilistic-based approach on small data regimes is clearly demonstrated in the experiments, where 3-12 samples per category per task are provided. Can the authors shed some light on the scenario when training with large amounts of data? Is varitionable-based approaches still favorable in that setting?


**Time Spent Reviewing:**

8

---

> ### Author Response · Authors · 2021-08-10
> **Response to Reviewer Q1H6**
>
> We thank Reviewer Q1H6 for stating one of the novelties of our method namely "it is the first time (for Gumbel-Softmax) being applied to the variational multi-task learning setup".
>
> 1. Thank you for the great suggestion. Compared to the deterministic baseline (BMTL), the training and inference time of our approach increase as the number of Monte Carlo (MC) samples is set higher, as shown in the following table (the unit for time is the second, experiments on Office-Home with the 5\% train-test split).
> In this paper, the number of MC samples is set to be 10, which is computationally efficient while yielding good performance (Table C15 in the supplementary). In this case, our method does cost extra training time but with 0.122s per iteration, which is still acceptable. When testing 1000 samples, our method only increases the extra 10\% test time of BMTL. Thus, our model doesn't cost much more time to surpass BMTL by 7.9\%.
> We will add the discussions and results in the paper.
> | Methods |   BMTL |  |VMTL |  | |
> | ---| ---| ---|---| ---| --- |
> | MC samples | $-$ | 1 | 10 | 50 | 100|
> |Training (per iteration)| 0.040 | 0.098 | 0.122 | 0.197 | 0.320 |
> |Inference (per 1000 test samples) | 0.325 | 0.343 | 0.357 | 0.371 | 0.426 |
> |
>
> 2. Thank you for those insightful comments.
>
>     (i) In the scenarios of training with large amounts of data, the knowledge transfer becomes less crucial since individual tasks suffer less from overfitting caused by limited data.
>
>     (ii) Yes, our variational-based approach is still favorable because it can produce comparable even better performance than baselines in the setting with large amounts of training data, as shown in Fig.2 of the paper (Up to 9,000 training samples).
> For your interest, we further show the results of using the 80% train-test split (more than 12,000 samples for training) in the following table (experiments on the *Office-Home*, *Office-Caltech* and *ImageCLEF* datasets), which we will add to the paper. Moreover, *Domainnet* is indeed a large-scale dataset, which provides a benchmark to demonstrate the effectiveness in large data regimes with approximately 0.6 million images distributed among 345 categories. The proposed methods achieve the best performance with more than 24,000 training samples in Table C14, which further demonstrates the effectiveness of our method when training with a large amount of data.
> |Methods| STL | VSTL | BMTL | VBMTL | VMTL-AC | VMTL |
> | --- | --- | --- | --- | --- | --- | --- |
> | Office-Home | 75.0$\pm$0.1 | 75.2$\pm$0.1 | 71.4$\pm$0.1 | 75.2$\pm$0.0 | $\underline{75.5\pm0.1}$ | $\textbf{75.9$\pm$0.1}$ |
> | Office-Caltech | 96.5$\pm$0.3 | 96.4$\pm$0.2 | 95.9$\pm$0.1 | $\underline{96.6\pm0.2}$ | $\underline{96.6\pm0.1}$ | $\textbf{96.9$\pm$0.3}$ |
> | ImageCLEF | 82.3$\pm$0.1 | 82.3$\pm$0.2 | 79.6$\pm$0.2 | 81.7$\pm$0.2 | $\underline{82.5\pm0.3}$ | $\textbf{82.7$\pm$0.2}$ |
> |
>
> **Limitations and societal impact:**
>
> **Q:** *The main limitation I can think of is the performance of the proposed variational Bayesian approach in the large data regime as mentioned above.*
>
> **A:** As indicated above this limitation is actually not there, improvements are just not that prominent. Our proposed methods show less significant improvements under the setting with amounts of training data than the setting with limited data. We note that in the large data regime, the knowledge transfer becomes less crucial since individual tasks suffer less from overfitting. Compared with other MTL models, our proposed methods still show certain improvement over individual tasks when provided with large amounts of training data. We will add those discussions about the limitation in the paper.
>
> **Q:** *I was wondering if there is a way to combine the advantages of the proposed approach with deterministic approaches so that it can generalize well in more settings?*
>
> **A:** We can in fact take advantages of deterministic approaches to generalize even better in more settings, e.g., large amounts of training data. In this case, we can train a large convolutional neural network fully end-to-end to extract more representative features specifically for individual tasks. This can be seamlessly combined with the advantages of our method in leveraging shared knowledge among tasks.  We will add this discussion in our paper. Thank you.

---

> > ### Comment · Reviewer_Q1H6 · 2021-08-28
> > **Post rebuttal**
> >
> > Thanks to the authors for providing the answers. I have no other concerns at this point. I will keep my original rating.

---

### Official Review · Reviewer_k1co · 2021-07-17

**Rating:** 6
**Confidence:** 4

**Summary:**

This paper proposes a variational multi-task learning method that uses Gumbel-Softmax priors for both the weight (w) of each task and the latent representation (z) of each data point. The Gumbel-Softmax trick is used for learning the relatedness between tasks. Amortized inference is also used for faster inference of the posteriors of the weight vectors and also for the latent representations. Experiments on different benchmark datasets show improvement over baselines especially in the low-data regime.

**Ethical Concerns:**

N.A.

**Limitations And Societal Impact:**

Yes. Maybe can also warn about possible negative transfer if tasks are not very related.

**Main Review:**

The paper is in general well-organized and easy to follow. The notations are clear and consistent throughout. Introducing a learnable Gumbel-Softmax prior for learning from relevant tasks is original and interesting. I like the extensive ablation studies on handling limited amounts of data, using gumbel-softmax prior, and amortization. Also the paper considers multiple baselines (both Bayesian and non-Bayesian) for comparison. In particular, the proposed method is able to perform relatively well with limited data.

My major remaining concern about the paper is about the use of different Gumbel-Softmax weights for the weight (w) and latent representation (z). Given that the weight and latent representations are coupled, maybe it makes more sense to have them to be the same?

I am also curious if there is any other suitable metric for comparing the uncertainty prediction of the Bayesian MTL methods. Currently it seems only accuracy or NSME is used as a metric for evaluation.


**Time Spent Reviewing:**

5 hours

---

> ### Author Response · Authors · 2021-08-10
> **Response to Reviewer k1co**
>
> We thank Reviewer k1co for stating our learnable Gumbel-Softmax prior is  ''original and interesting'' the notations in the paper are ''clear and consistent throughout'' and the paper is ''well-organized and easy to follow''.
>
> **Q:** *My major remaining concern about the paper is about the use of different Gumbel-Softmax weights for the weight (w) and latent representation (z). Given that the weight and latent representations are coupled, maybe it makes more sense to have them to be the same?*
>
> **A:** We apologize for not making this clear enough.
> We provide $\mathbf{w}$ and $\mathbf{z}$ with different weights, motivated by the fact that latent variables are different in terms of capturing uncertainty: $\mathbf{w}$ is at the category level while $\mathbf{z}$ at the instance level.
> Moreover, we conduct experiments using the same Gumbel-Softmax weights for both variables on the *Office-Home* dataset. As shown in the following table, using different weights is slightly better than using the same Gumbel-Softmax weights. We will add this to the paper.
>
> | Train-test split |   5\%  |  10\%  | 20\% |
> | --- | ---| --- | --- |
> | Same weights | 58.2$\pm$0.1 | 64.8$\pm$0.1 | 68.7$\pm$0.3 |
> | Different weights |  $\textbf{58.3$\pm$0.1}$ | $\textbf{65.0$\pm$0.0}$ | $\textbf{69.2$\pm$0.2}$ |
> |
>
> **Q:** *I am also curious if there is any other suitable metric for comparing the uncertainty prediction of the Bayesian MTL methods. Currently it seems only accuracy or NSME is used as a metric for evaluation.*
>
> **A:** Thank you for this constructive comment. For Bayesian methods, it is indeed necessary to quantify the model's ability to handle uncertainty. We looked into related references and didn't find such a measure for comparing the uncertainty prediction. Thus, we adopt a new metric for evaluating the uncertainty prediction, the ratio of the average entropy of failure cases and properly classified samples.  If the ratio is higher, the Bayesian methods predict failure cases with more uncertainty and predict successful cases with more confidence. As shown in the following table (higher is better, experiments on *Office-Home*), VMTL has higher entropy ratios, which demonstrates the effectiveness of our model to handle the uncertainty. We will add the definition of this metric and the results in the paper.
>
> | Train-test split |   5\%  |  10\%  | 20\% |
> | ---| ---| ---|---|
> | Bakker et al. [3] | 2.469 | 2.625 | 3.031 |
> | VBMTL | 4.111 | 4.430 | 5.460 |
> | VMTL | $\textbf{4.546}$ | $\textbf{4.472}$ | $\textbf{5.584}$ |
> |
>
> **Limitations and societal impact:**
>
> Indeed, one should be aware of possible negative transfer among tasks and take this into account during the model design for multi-task learning. We should have discussed this more clearly in our paper. Our method handles possible negative transfer effectively through the design of the Gumbel-Softmax priors. The Gumbel-Softmax technique encourages the model to reduce the interference from the not-very-related tasks by minimizing the corresponding mixing weights. The more negative the effects of interference between pairwise tasks are, the smaller the mixing weight is likely to be.
> We will make this clearer in our paper. Thank you.

---

> > ### Comment · Reviewer_k1co · 2021-08-30
> > **After authors' response**
> >
> > Thank the authors for responding to my questions. I will keep my original rating.

---

### Author Response · Authors · 2021-08-20
**Look forward to your feedback**

Dear reviewers,


We truly thank you for spending your time reading our paper and the rebuttals. As the discussion period is drawing to a close, we look forward to hearing your feedback about whether we have resolved your concerns in the rebuttals. We would like to discuss if you still have any concerns. Thank you!


Best wishes,

Authors.

---

### Decision · Program_Chairs · 2021-09-27

**Decision:**

Accept (Poster)

**Comment:**

The reviewers all found the approach to be “original and interesting,” with their main concerns addressed in the discussion. The technique looks to be very useful in the multi-task setting with limited data. During the discussion period, additional experiments showed that it doesn’t hurt in the large-data setting either, although the improvement is less pronounced. To me, this shows that the Bayesian approach seems to be doing the right thing: not getting in the way of single-task performance when enough data is available, but leveraging multiple tasks when data is scarce. One reviewer has concerns that the approach is too complex to be widely adopted, but given that code was made available as part of the submission, I am less concerned about this.